# The ortholog of human REEP1-4 is required for autophagosomal enclosure of ER-phagy/nucleophagy cargos in fission yeast

**Chen-Xi Zou**[1,2], **Zhu-Hui Ma**[1], **Zhao-Di Jiang**[1], **Zhao-Qian Pan**[1], **Dan-Dan Xu**[1], **Fang Suo**[1], **Guang-Can Shao**[1], **Meng-Qiu Dong**[1,3], **Li-Lin Du**[1,3]*

1 National Institute of Biological Sciences, Beijing, China, 2 College of Life Sciences, Beijing Normal University, Beijing, China, 3 Tsinghua Institute of Multidisciplinary Biomedical Research, Tsinghua University, Beijing, China

* dulilin@nibs.ac.cn

**Data Availability Statement:** All relevant data are within the paper and its Supporting Information files.

## Abstract

Selective macroautophagy of the endoplasmic reticulum (ER) and the nucleus, known as ER-phagy and nucleophagy, respectively, are processes whose mechanisms remain inadequately understood. Through an imaging-based screen, we find that in the fission yeast *Schizosaccharomyces pombe*, Yep1 (also known as Hva22 or Rop1), the ortholog of human REEP1-4, is essential for ER-phagy and nucleophagy but not for bulk autophagy. In the absence of Yep1, the initial phase of ER-phagy and nucleophagy proceeds normally, with the ER-phagy/nucleophagy receptor Epr1 coassembling with Atg8. However, ER-phagy/nucleophagy cargos fail to reach the vacuole. Instead, nucleus- and cortical-ER-derived membrane structures not enclosed within autophagosomes accumulate in the cytoplasm. Intriguingly, the outer membranes of nucleus-derived structures remain continuous with the nuclear envelope-ER network, suggesting a possible outer membrane fission defect during cargo separation from source compartments. We find that the ER-phagy role of Yep1 relies on its abilities to self-interact and shape membranes and requires its C-terminal amphipathic helices. Moreover, we show that human REEP1-4 and budding yeast Atg40 can functionally substitute for Yep1 in ER-phagy, and Atg40 is a divergent ortholog of Yep1 and REEP1-4. Our findings uncover an unexpected mechanism governing the autophagosomal enclosure of ER-phagy/nucleophagy cargos and shed new light on the functions and evolution of REEP family proteins.

## Introduction

In eukaryotes, the endoplasmic reticulum (ER) is an intricate membrane organelle composed of interconnected sheet-like structures and tubular networks [1,2]. The formation and maintenance of ER morphology involve 2 conserved families of integral membrane proteins, the reticulons (RTNs) and the REEP family proteins [3,4]. The ER plays a crucial role in many cellular processes, such as protein folding, lipid synthesis, ion homeostasis, and communication with

**Funding:** This work was supported by intramural funding from the National Institute of Biological Sciences, Beijing, and the Tsinghua Institute of Multidisciplinary Biomedical Research, Tsinghua University (LLD). The funders had no roles in study design, data collection and analysis, decision to publish, or manuscript preparation.

**Competing interests:** The authors have declared that no competing interests exist.

**Abbreviations:** AIM, Atg8-interacting motif; AID, auxin-inducible degron; APH, amphipathic helix; BiFC, bimolecular fluorescence complementation; ER, endoplasmic reticulum; FRAP, fluorescence recovery after photobleaching; RTN, reticulon; TCA, trichloroacetic acid; TEM, transmission electron microscopy; TMH, transmembrane helix.

other organelles [5]. Disturbances of ER functions have been implicated in a wide range of human diseases [6].

Under starvation and ER stress conditions, portions of the ER are turned over through macroautophagy (hereafter autophagy), in a process termed "ER-phagy." During ER-phagy, ER membrane fragments are sequestered into autophagosomes, which are double-membrane vesicles that deliver cargos to the lysosome/vacuole for degradation [7–9]. In yeasts, the ER mainly consists of the nuclear envelope and the cortical ER [10,11]. Both subcompartments of the ER can be targeted by ER-phagy. The autophagic sequestration of the nuclear envelope may result in the engulfment of intranuclear components into autophagosomes. Thus, ER-phagy and nucleophagy may occur concurrently [12,13].

The recruitment of the autophagic machinery during ER-phagy and nucleophagy is mediated by specialized autophagy receptors. In recent years, a large number of ER-phagy receptors have been identified, including FAM134B, FAM134A, FAM134C, SEC62, RTN3L, CCPG1, ATL3, TEX264, p62, CALCOCO1, and C53 in mammals [14–25], Atg39 and Atg40 in budding yeast [12], and Epr1 in fission yeast [26]. These ER-phagy receptors, which are integral or peripheral ER membrane proteins, all harbor binding motifs for Atg8 family proteins and, consequently, can mediate the association between the ER and Atg8-decorated autophagic membranes. Some of them solely promote the ER-Atg8 connection during ER-phagy. For example, in the fission yeast *Schizosaccharomyces pombe*, the soluble ER-phagy receptor Epr1, which localizes to the ER through binding integral ER membrane proteins VAPs, can be rendered dispensable by fusing an Atg8-interacting motif (AIM) to an integral ER membrane protein Erg11 [26]. On the other hand, mammalian FAM134B and budding yeast Atg40, both of which are integral membrane proteins, have roles beyond establishing the ER-Atg8 connection [27–31].

In budding yeast and mammalian cells, genetic screenings have uncovered a large number of genes important for ER-phagy [32,33], suggesting that ER-phagy receptors are not the only factors specifically required for ER-phagy. We conduct an imaging-based chemical mutagenesis screen in fission yeast and identify Yep1, the ortholog of human REEP1-4, as a crucial factor in both ER-phagy and nucleophagy. Yep1 is not required for ER-phagy/nucleophagy initiation but is needed for autophagosomal enclosure of cargo membrane structures. The ER-phagy function of Yep1 requires not only its first 113 residues that can self-interact and shape ER membrane but also its C-terminal amphipathic helices, which are dispensable for the membrane-shaping ability. Interestingly, the ER-phagy function of Yep1 can be substituted by human REEP1-4 and budding yeast Atg40. Phylogenetic analysis suggests that Atg40 is a divergent ortholog of Yep1 and REEP1-4. We propose that Yep1 and its equivalents in other eukaryotes play a crucial but previously unanticipated role in ER-phagy and nucleophagy.

## Results

### Yep1 is required for ER-phagy and nucleophagy

To identify *S. pombe* genes important for ER-phagy, we performed an imaging-based mutant screen (**Fig 1A**). Chemical mutagenesis was applied to a strain in which the copy numbers of 23 general autophagy genes were doubled to reduce the chance of isolating their mutants in the screen. ER-phagy was monitored by examining microscopically the relocalization of the integral ER membrane protein Ost4-GFP to the vacuole upon DTT treatment [26,34,35]. A mutant clone isolated in this screen was subjected to next-generation sequencing-assisted bulk segregant analysis [36], which indicated that a missense mutation (T17M) in an uncharacterized gene *SPBC30D10.09c* is a candidate causal mutation (**S1A Fig**). For reasons described below, we named this gene *yep1* (for Yop1- and REEP-related protein required for ER-phagy). Yep1 protein belongs to the REEP protein family. This protein family encompasses 2

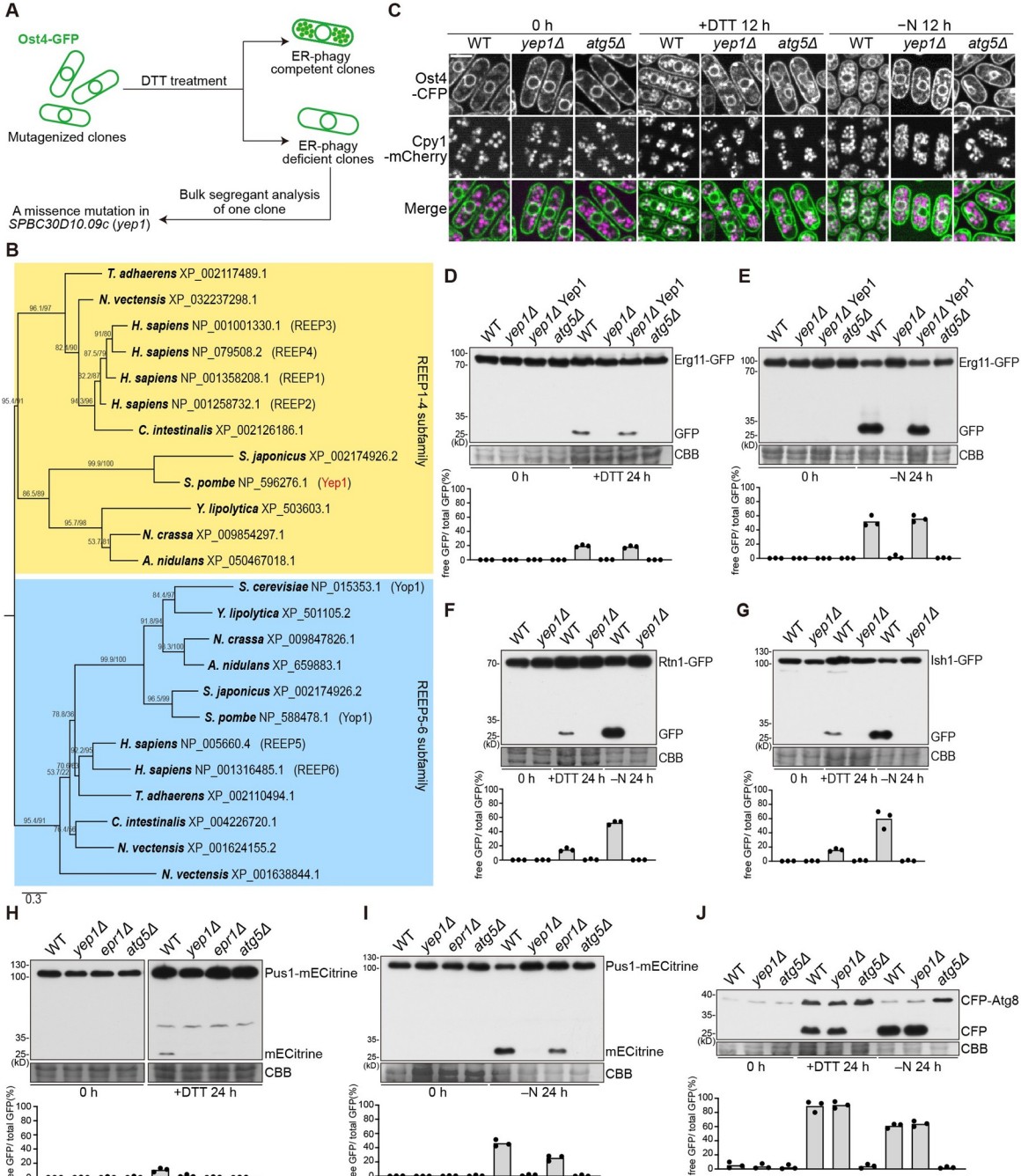

**Fig 1. Yep1 is required for ER-phagy and nucleophagy. (A)** Schematic of the imaging-based screen for ER-phagy mutants. Mutagenized clones harboring the Ost4-GFP reporter were treated with DTT for 16 hours to induce ER-phagy. The phenotype-causing mutation in an ER-phagy-deficient clone was identified by bulk segregant analysis. **(B)** Phylogenetic relationships of REEP family proteins in representative metazoan and fungal species. A maximum likelihood tree was constructed using IQ-TREE and rooted by midpoint rooting. Branch labels are the SH-aLRT support values (%) and the UFBoot support values (%) calculated by IQ-TREE. Scale bar indicates 0.3 substitutions per site. **(C)** Relocalization of the integral ER membrane protein Ost4-CFP to the vacuole after nitrogen starvation (−N) or DTT treatment (+DTT) was abolished in *yep1Δ* cells. Wild-type (WT), *yep1Δ*, and *atg5Δ* cells expressing Ost4-CFP were examined by microscopy before and after 12-hour starvation or DTT treatment. Cpy1-mCherry is a vacuole lumen marker. Bar, 5 μm. **(D–E)** Autophagic processing of the ER membrane protein Erg11-GFP was abolished in *yep1Δ* cells. Cells expressing Erg11-GFP were collected before and after 24-hour starvation **(D)** and DTT treatment **(E)**, and total lysates were analyzed by immunoblotting using an antibody against GFP. Post-immunoblotting staining of the PVDF membrane using Coomassie Brilliant Blue (CBB) served as the loading control. The third sample (*yep1Δ* Yep1) is a *yep1Δ* strain transformed with an integrating plasmid expressing Yep1 tagged with mCherry. The blot image is a representative of triplicate experiments. Quantitation of triplicate experiments is shown below the image.

**(F)** Autophagic processing of the cortical ER protein Rtn1-GFP was abolished in *yep1Δ* cells. The blot image is a representative of triplicate experiments. Quantitation of triplicate experiments is shown below the image. **(G)** Autophagic processing of the nuclear envelope protein Ish1-GFP was abolished in *yep1Δ* cells. The blot image is a representative of triplicate experiments. Quantitation of triplicate experiments is shown below the image. **(H)** DTT-induced autophagic processing of the nucleoplasmic protein Pus1-mECtrine was abolished in *epr1Δ* and *yep1Δ* cells. The blot images are representatives of triplicate experiments. Quantitation of triplicate experiments is shown below the images. **(I)** Nitrogen starvation-induced autophagic processing of the nucleus protein Pus1-mECtrine was diminished in *epr1Δ* cells and abolished in *yep1Δ* cells. The blot image is a representative of triplicate experiments. Quantitation of triplicate experiments is shown below the image. **(J)** Autophagic processing of CFP-Atg8 was largely normal in *yep1Δ* cells. The blot image is a representative of triplicate experiments. Quantitation of triplicate experiments is shown below the image. Numerical data underlying panels D-J can be found in S1 Data, and raw images for panels D-J can be found in S1 Raw Images.

subfamilies, both of which are present in most metazoan and fungal species (**Fig 1B**) [37–40]. There are 6 REEP family proteins in humans. Among them, REEP1-4 proteins belong to 1 subfamily and REEP5-6 proteins belong to the other. *S. pombe* has 2 REEP family proteins, Yep1 and Yop1. Yep1 is the ortholog of human REEP1-4, and Yop1 is the ortholog of human REEP5-6 (**Fig 1B**). Human REEP1-4 are ER-localizing proteins [38,41,42], Similarly, we found that Yep1 exhibited an ER localization pattern during vegetative growth (**S1B Fig**).

Threonine 17 in Yep1 is a conserved residue in REEP1-4 subfamily proteins. Thus, we hypothesized that the T17M mutation may compromise the function of Yep1 and consequently cause an ER-phagy defect. Consistent with this idea, deletion of *yep1* severely diminished the relocalization of Ost4-CFP to the vacuole upon DTT treatment (+DTT) or nitrogen starvation treatment (−N) (**Fig 1C**). *yep1Δ* cells also exhibited a severe defect in the autophagic processing of GFP-tagged integral ER membrane protein Erg11 into free GFP (**Fig 1D and 1E**). Reintroducing Yep1 into *yep1Δ* cells completely rescued the defect. These results indicate that Yep1 is essential for ER-phagy.

As Ost4 and Erg11 localize to both the cortical ER and the nuclear envelope, we also examined the autophagic processing of Rtn1-GFP, which localizes exclusively at the cortical ER, and Ish1-GFP, which is a nuclear envelope protein. The loss of Yep1 abolished DTT- and starvation-induced processing of Rtn1-GFP and Ish1-GFP (**Fig 1F and 1G**), indicating that Yep1 is required for the autophagy of both subcompartments of the ER.

The observation that the nuclear envelope protein Ish1-GFP is subjected to autophagy suggested to us that the autophagic turnover of nucleoplasmic components, i.e., nucleophagy, may occur in *S. pombe*. Indeed, under both DTT and starvation treatments, a nucleoplasmic protein Pus1-mECitrine was processed to free mECitrine in an Atg5-dependent manner (**Fig 1H and 1I**). The loss of the ER-phagy receptor Epr1 diminished the processing of Pus1-mECitrine, suggesting that Epr1 also acts as a nucleophagy receptor. Deletion of *yep1* abolished the processing of Pus1-mECitrine in both DTT- and starvation-treated cells, indicating that Yep1 is essential for nucleophagy. Consistent with its role in ER-phagy and nucleophagy, Yep1 was observed to form puncta at approximately 30% of the sites where both Atg8 and Epr1 formed puncta (**S1C and S1D Fig**), suggesting that Yep1 participates in ER-phagy and nucleophagy at sites of autophagosome assembly.

In contrast to the severe ER-phagy and nucleophagy defects of *yep1Δ* cells, bulk autophagy in *yep1Δ* cells was largely normal as indicated by the processing of CFP-Atg8 (**Fig 1J**). In addition, another readout of bulk autophagy, the processing of fluorescent protein-tagged cytosolic protein Tdh1 (glyceraldehyde-3-phosphate dehydrogenase (GAPDH)) [43], was also largely unaffected in *yep1Δ* cells (**S1E Fig**). Consistent with the lack of bulk autophagy defects, transmission electron microscopy (TEM) analysis showed that autophagosome accumulation in the *fsc1Δ* mutant, which is defective in autophagosome–vacuole fusion [44], was not notably affected by the deletion of *yep1* (**S1F Fig**). Inspection of the electron micrographs showed that in *fsc1Δ* cells, there are autophagosomes containing a ring-shaped membrane structure,

possibly of the ER/nuclear envelope origin. We call such autophagosomes double-ring structures. The level of double-ring structures was higher under DTT treatment than under starvation treatment (**S1G Fig**). This is inconsistent with the results of the fluorescent protein cleavage assay, possibly because the cleavage assay underestimates autophagic flux under DTT treatment due to an inhibition of vacuolar proteolysis by DTT treatment [35]. The number of double-ring structures per cell was markedly lower in *fsc1Δ yep1Δ* cells than in *fsc1Δ* cells (**S1F and S1G Fig**), suggesting a defect in forming autophagosomes containing ER-phagy and nucleophagy-related cargo membrane structures.

## Yep1 acts independently of Epr1 and is not required for the initiation of ER-phagy

The most well-understood type of proteins important for ER-phagy but not bulk autophagy are ER-phagy receptors, whose functions rely on their interactions with Atg8. We examined whether Yep1 can interact with Atg8 using the Pil1 co-tethering assay [45]. Epr1, but not Yep1, interacted with Atg8 in this assay (**S1H Fig**), indicating that Yep1 is unlikely to be an ER-phagy receptor.

As the proper functioning of the fission yeast ER-phagy receptor Epr1 depends on Ire1, which up-regulates the expression of Epr1 during ER stress [26], we examined the possibility that Yep1 also promotes the expression of Epr1. Immunoblotting analysis showed that ER stress-induced up-regulation of the protein level of Epr1 occurred normally in *yep1Δ* cells (**S1I Fig**), ruling out this possibility. No interaction between Yep1 and Epr1 was detected using a yeast two-hybrid assay (**S1J Fig**). Furthermore, even though the role of Epr1 can be bypassed by fusing an artificial AIM to the integral ER membrane protein Erg11 [26] (**S1K Fig**), this fusion did not suppress the ER-phagy defect of *yep1Δ* (**S1L Fig**), indicating that the main role of Yep1 is not promoting the function of Epr1. Consistent with this idea, increasing the level of Epr1 did not suppress *yep1Δ* (**S1L Fig**), and increasing the level of Yep1 did not suppress *epr1Δ* (**S1K Fig**). Together, these results suggest that Yep1 and Epr1 play different roles in ER-phagy.

Next, we examined whether *yep1Δ* affects the localization of Epr1 and Atg8 during ER-phagy. In both wild-type and *yep1Δ* cells, Epr1 and Atg8 colocalized at punctate structures shortly after ER-phagy induction by DTT or starvation treatment (**S2A Fig**), indicating that Yep1 is not essential for the initial stage of ER-phagy during which Epr1 mediates a connection between the ER and Atg8-decorated autophagic membranes.

## Yep1 is required for the autophagosomal enclosure of ER-phagy and nucleophagy cargos

In our analysis of nucleophagy phenotypes using strains expressing the nucleoplasmic protein Pus1-mECitrine, we noticed that Pus1-mECitrine formed cytoplasmic puncta in *yep1Δ* cells after ER-phagy induction (**Fig 2A**). We then examined an inner nuclear membrane protein Bqt4 and found that it also formed cytoplasmic puncta in *yep1Δ* cells after ER-phagy induction (**Fig 2B**). No Pus1 or Bqt4 puncta were observed in *atg5Δ* cells and *yep1Δ atg5Δ* cells (**Fig 2A and 2B**), suggesting that these puncta are autophagy-related structures. Under starvation treatment, no Pus1 or Bqt4 puncta were observed in wild-type cells. Under DTT treatment, a smaller number of Pus1 and Bqt4 puncta were observed in wild-type cells, possibly because DTT treatment causes a mild accumulation of autophagosomes in the cytoplasm (see below). We examined whether Pus1 puncta and Bqt4 puncta in *yep1Δ* cells colocalize and found that Pus1 puncta almost always overlapped with Bqt4 puncta and a great majority (>80%) of Bqt4 puncta overlapped with Pus1 puncta (**Figs 2C and S2B**). Thus, the nucleophagy defect of *yep1Δ* is accompanied by the cytoplasmic accumulation of nucleus-derived membrane structures containing

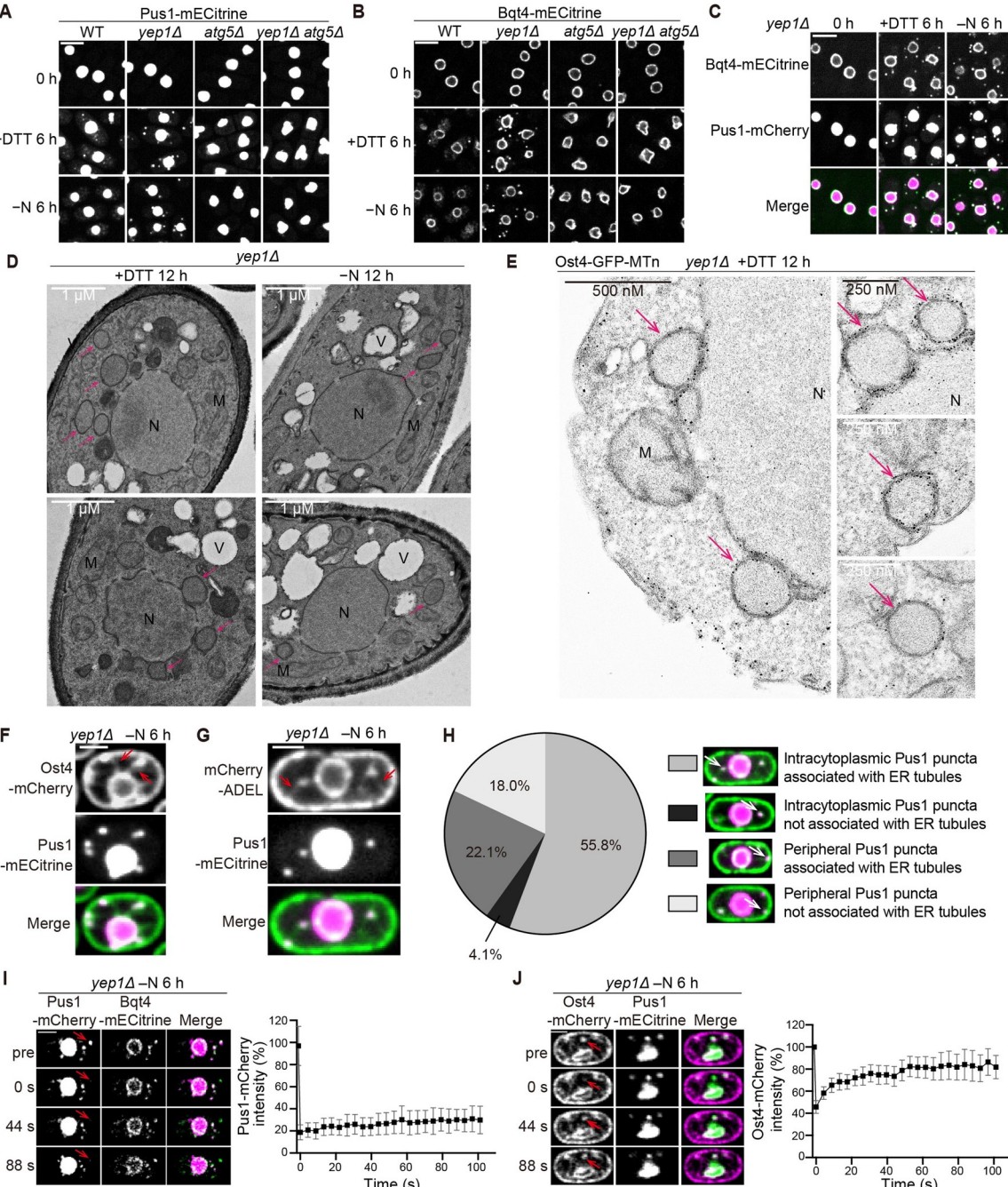

**Fig 2. Yep1 is required for the autophagosomal enclosure of ER-phagy and nucleophagy cargos. (A)** Bqt4-mECtrine formed cytoplasmic puncta in *yep1Δ* cells after nitrogen starvation (−N) or DTT treatment. Bar, 5 μm. **(B)** Pus1-mECtrine formed cytoplasmic puncta in *yep1Δ* cells after nitrogen starvation or DTT treatment. Bar, 5 μm. **(C)** The colocalization between Bqt4 puncta and Pus1 puncta accumulating in *yep1Δ* cells. Bar, 5 μm. **(D)** Electron microscopy analysis of *yep1Δ* cells treated with DTT or nitrogen starvation. N, nucleus; V, vacuole; M, mitochondrion. Ring-shaped membrane structures are denoted by pink arrows. Bar, 1 μm. **(E)** Electron microscopy analysis of gold nanoparticle-labeled membrane structures of the ER/nuclear envelope origin in *yep1Δ* cells. MTn tagging of Ost4 allowed membrane structures of the ER/nuclear envelope origin to be labeled by EM-visible gold nanoparticles. N, nucleus; M, mitochondrion. Gold nanoparticle-labeled ring-shaped membrane structures are denoted by pink arrows. **(F and G)** Pus1 puncta that accumulated in nitrogen starvation–treated *yep1Δ* cells colocalized with puncta formed by ER markers Ost4-mCherry (**F**) and mCherry-ADEL (**G**), and these ER marker puncta were often associated with ER tubules. Red arrows denote puncta-associated ER tubules. Bar, 2 μm. **(H)** Percentages of different types cytoplasmic Pus1 puncta in images acquired in the analysis shown in G (more than 200 puncta were examined). Pus1 puncta were classified into 4 types based on their locations (intracytoplasmic or peripheral) and whether the colocalizing mCherry-ADEL punctum was associated with ER tubules. **(I and J)** FRAP analysis of cytoplasmic Pus1-mCherry puncta (**I**)

and Pus1-colocalizing Ost4-mCherry puncta (**J**) in *yep1Δ* cells treated with nitrogen starvation. The images were taken before (pre) and after photobleaching. Photobleached puncta are indicated by red arrows. Bar, 2 μm. Fluorescence intensities of the puncta were quantitated and are shown as means ± standard deviations (*n* = 31 for Pus1-mCherry puncta and *n* = 30 for Ost4-mCherry puncta). Numerical data underlying panels I and J can be found in S1 Data.

inner nuclear membrane components and nucleoplasmic components. In *yep1Δ* cells, less than 10% of Pus1 puncta and Bqt4 puncta colocalized with Atg8 puncta (**S2C and S2D Fig**), indicating that most nucleus-derived membrane structures accumulating in the cytoplasm of *yep1Δ* cells are not associated with autophagic membranes decorated with Atg8.

To better understand the nature of the structures represented by cytoplasmic puncta of Bqt4 and Pus1, we performed TEM analysis. This analysis showed that ring-shaped membrane structures accumulated in the cytoplasm of *yep1Δ* cells but not *yep1Δ atg5Δ* cells (**Figs 2D and S2E and S2F**). These structures differ in location from the ring-shaped autophagosomes that accumulated in the *fsc1Δ* cells (**S1D Fig**). Autophagosomes accumulating in the *fsc1Δ* cells are almost always juxtaposed to vacuoles. In contrast, ring-shaped membrane structures in the cytoplasm of *yep1Δ* cells were never observed juxtaposed to vacuoles. The shape and size of these structures are similar to those of the inner rings in the double-ring structures observed in the *fsc1Δ* cells (**S2G Fig**). Thus, we proposed that these structures are ER-phagy/nucleophagy cargos not enclosed within autophagosomes, and they include structures represented by the Bqt4 and Pus1 puncta. In DTT-treated, but not starvation-treated wild-type cells, autophagosomes juxtaposed to vacuoles can be observed (**S2E Fig**). They likely include structures corresponding to the Bqt4 and Pus1 puncta observed in wild-type cells by fluorescence microscopy.

To further examine the nature of the ring-shaped membrane structures accumulating in the cytoplasm of *yep1Δ* cells, we fused a genetically encoded EM tag, MTn, to the integral ER membrane protein Ost4 and examined its distribution in *yep1Δ* cells using a recently developed EM technology [46]. MTn-generated gold nanoparticles were observed not only on the nuclear envelope and the cortical ER but also on ring-shaped cytoplasmic membrane structures resembling the ring-shaped structures observed in the TEM analysis (**Fig 2E**), confirming that nucleus- and/or cortical-ER-derived structures without surrounding autophagic membranes accumulate in the cytoplasm of *yep1Δ* cells.

To obtain more corroborating evidence on the accumulation of ER-phagy/nucleophagy cargos in *yep1Δ* cells, we employed the degron protection assay [47]. In this assay, an auxin-inducible degron (AID) tag is fused to a protein so that the protein is degraded when exposed to the cytosol but not when residing inside a membrane compartment. We used the cytosolic protein Pyk1, a bulk autophagy cargo, to verify whether a protein enclosed within the autophagosome is resistant to AID-mediated degradation. In untreated cells, Pyk1-AID-mECitrine was completely degraded upon the addition of the auxin analog 5-adamantyl-IAA (Ad-IAA) [48] (**S3A Fig**). When autophagy was induced by nitrogen starvation, Pyk1-AID-mECitrine signal in small round-shaped structures, which are vacuoles labelled by the vacuole lumen marker Cpy1-mCherry, persisted in wild-type cells after the addition of Ad-IAA. Similarly, persisting Pyk1-AID-mECitrine signal in vacuoles was observed in nitrogen-starved *yep1Δ* cells, confirming that bulk autophagy is largely normal in *yep1Δ* cells. In *fsc1Δ* cells, which are defective in autophagosome–vacuole fusion, persisting Pyk1-AID-mECitrine signal in structures not overlapping with vacuoles was observed after the addition of Ad-IAA. These structures are presumably autophagosomes, suggesting that AID-mediated degradation does not happen to proteins enclosed within autophagosomes. For our analysis of ER-phagy/nucleophagy cargos accumulating in *yep1Δ* cells, we chose Epr1 as the degradation target as it is a peripheral membrane protein facing the cytosol and is concentrated at the sites of ER-phagy/

nucleophagy. As expected, in untreated wild-type or *yep1Δ* cells, signals of Epr1-AID-mECitrine completely disappeared after the addition of Ad-IAA (S3B Fig). In contrast, in wild-type cells starved for 6 hours, Epr1-AID-mECitrine relocalized to the vacuole lumen, and the vacuolar signal persisted after the addition of Ad-IAA. In *yep1Δ* cells, Epr1-AID-mECitrine formed cytoplasmic puncta, some of which colocalized with Pus1 cytoplasmic puncta. Upon the addition of Ad-IAA, Epr1-AID-mECitrine puncta completely disappeared (S3B Fig), indicating that in *yep1Δ* cells, the outer surface of nucleus-derived membrane structures accumulating in the cytoplasm is exposed to the cytosol.

To determine whether cortical-ER-derived membrane structures not enclosed within autophagosomes also accumulated in *yep1Δ* cells, we applied the degron protection assay to the integral ER membrane protein Rtn1, which localizes to the cortical ER but not the nuclear envelope (S3C Fig). Rtn1-AID-mECitrine signal in untreated wild-type and *yep1Δ* cells disappeared upon the addition of Ad-IAA. In starvation-treated wild-type cells, Rtn1-AID-mECitrine partially relocalized to the vacuole lumen and the vacuole-localized signal persisted upon the addition of Ad-IAA. In starvation-treated *yep1Δ* cells, Rtn1-AID-mECitrine formed cytoplasmic puncta, which disappeared after the addition of Ad-IAA (S3C Fig), suggesting that like the situation of nucleophagy cargos, cortical-ER-phagy cargos not enclosed within autophagosomes also accumulated in *yep1Δ* cells.

We next investigated whether the ER-phagy/nucleophagy cargo structures that accumulated in *yep1Δ* cells were fully separated from the source compartments. Live cell imaging showed that in starvation-treated *yep1Δ* cells, cytoplasmic puncta formed by the nucleoplasmic protein Pus1-mECitrine colocalized with puncta formed by the ER membrane marker Ost4-mCherry and the ER lumen marker mCherry-ADEL (ADEL is an ER retention signal) (Fig 2F and 2G). Notably, these Pus1-positive ER marker puncta were often associated with cytoplasmic ER tubules or cortical ER (Fig 2F and 2G and 2H), suggesting that the nucleophagy cargo structures remained attached to the nuclear envelope/ER network. However, the signals of Pus1 and the inner nuclear membrane protein Bqt4 were never observed on ER tubule-like structures (Fig 2A–2C and 2F and 2G), indicating that the inner contents and the inner membranes of the nucleophagy cargo structures were no longer continuous with the nucleoplasm and the inner nuclear membrane, respectively. Supporting this, fluorescence recovery after photobleaching (FRAP) analysis showed that the fluorescence signals of photobleached Pus1-mCherry puncta only recovered slightly (Fig 2I). The minor increase in fluorescence signals can perhaps be attributed to the reversible photoswitching of mCherry [49]. In contrast, the fluorescence signals of Pus1-positive Ost4-mCherry puncta substantially recovered after photobleaching (**Figs 2H and 2I and S3F**), supporting that the outer membranes of the nucleophagy cargo structures were continuous with the nuclear envelope/ER network. Using TEM analysis, we also observed nucleus- and/or cortical-ER-derived cargo structures that had filamentous membrane protrusions, which are likely ER tubules (S3D Fig).

Taken together, the above findings demonstrate that Yep1 is required for the autophagosomal enclosure of ER-phagy/nucleophagy cargos. In the absence of Yep1, ER-phagy/nucleophagy cargo structures devoid of surrounding autophagosomal membranes accumulate in the cytoplasm. The inner membranes of these cargo structures are fully disconnected from the source compartments. However, the outer membranes of these cargo structures remain continuous with the nuclear envelope/ER network.

## Yep1 possesses the ability to shape the ER membrane

In fission yeast, 3 ER-shaping proteins, the REEP family protein Yop1, the RTN family protein Rtn1, and the TMEM33 family protein Tts1, localize to tubular ER and act in a partially

redundant manner to maintain tubular ER [50]. We observed that Yep1 exhibited colocalization with Rtn1, Yop1, and Tts1 (**Figs 3A and S4A**), suggesting that Yep1 may share similar functions with these proteins.

In the absence of Rtn1, Yop1, and Tts1, the cortical ER becomes less reticulate and more sheet-like, with the frequent appearance of large holes in images of the top or bottom plane of the cells and extended gaps in images of the midplane of the cells [50]. We found that this alteration of ER morphology can be reversed to a large extent by either introducing back Rtn1 or increasing the expression level of Yep1 (**Figs 3B and S4B**). Thus, Yep1, when overexpressed, can fulfill the function of maintaining tubular ER independently of Rtn1, Yop1, and Tts1.

The ER structure aberration caused by the loss of Rtn1, Yop1, and Tts1 also leads indirectly to a severe septum positioning defect, manifesting as long-axis septum, multiple septa, and tilted septum [50]. This phenotype is easier to score than the ER morphology phenotype and can reveal the weak phenotypes of single deletion mutants lacking Rtn1, Yop1, or Tts1 (**Fig 3C**). We observed that *yep1Δ* caused a noticeable but even weaker septum positioning defect than *rtn1Δ*, *yop1Δ*, or *tts1Δ*. Combined *yep1Δ* with the double and triple deletion of *rtn1*, *yop1*, and *tts1* invariably resulted in a more severe phenotype (**Fig 3C**). The most pronounced phenotypic enhancement was observed when *yep1Δ* was combined with the *rtn1Δ tts1Δ* double deletion. The defect of *rtn1Δ tts1Δ yep1Δ* can be rescued to the level of *rtn1Δ tts1Δ* by reintroducing Yep1 (**S4C Fig**). Moreover, increasing the expression level of Yep1 ameliorated the septum positioning defect of *rtn1Δ yop1Δ tts1Δ* (**S4D Fig**). These results demonstrate that Yep1 shares the membrane-shaping ability of Rtn1, Yop1, and Tts1 and contributes to the maintenance of normal ER structure.

We assessed whether Rtn1, Yop1, and Tts1 function in ER-phagy. DTT and starvation-induced processing of Erg11-GFP was only slightly diminished in the *rtn1Δ yop1Δ tts1Δ* triple deletion mutant (**S4E Fig**), suggesting that ER-phagy still occurs in the absence of these 3 proteins. In addition, overexpression of Rtn1, Yop1, or Tts1 did not alleviate the severe ER-phagy defect of *yep1Δ* (**S4F Fig**). Thus, Rtn1, Yop1, and Tts1 cannot substitute for the essential role of Yep1 in ER-phagy.

## Yep1 self-interaction is important for its membrane-shaping ability and ER-phagy function

Several ER-shaping proteins, including Yop1, are known to self-interact [3,4,51–53]. It has been recently shown that the formation of curved-shape homo-oligomer of ER-shaping proteins is responsible for generating the tubular membrane shape [54]. Using the co-immunoprecipitation assay, we found that Yep1 can self-interact (**Fig 3D**). Bimolecular fluorescence complementation (BiFC) confirmed the self-interaction of Yep1 and suggested that Yep1 undergoes self-interaction on the ER membrane (**S5A Fig**).

We used AlphaFold-Multimer to predict the structures of Yep1 homo-oligomers [55] (**Figs 3E and S5B**). Regardless of the number of Yep1 sequences (from 2 to 8) in the input, AlphaFold-Multimer only predicted one type of oligomeric structure—the structure of the Yep1 dimer, indicating that the dimer is the preferred oligomerization state of Yep1. In the predicted structure of the Yep1 dimer (**Fig 3F**), within each Yep1 molecule, there are 3 long α-helices in the N-terminal region. They largely encompass the 3 transmembrane segments predicted by TOPCONS (**Figs 3G and S5C**). The C-terminal cytoplasmic region contains 2 short α-helices, 2 long α-helices, and a disordered tail [56] (**Figs 3G and S5C**). Intermolecular contacts mainly involve the first 2 transmembrane helices (TMHs). The 2 short C-terminal helices (amino acids 98–113) also contribute to the dimer interface by engaging the N-terminal helices of the other molecule (**Fig 3F**).

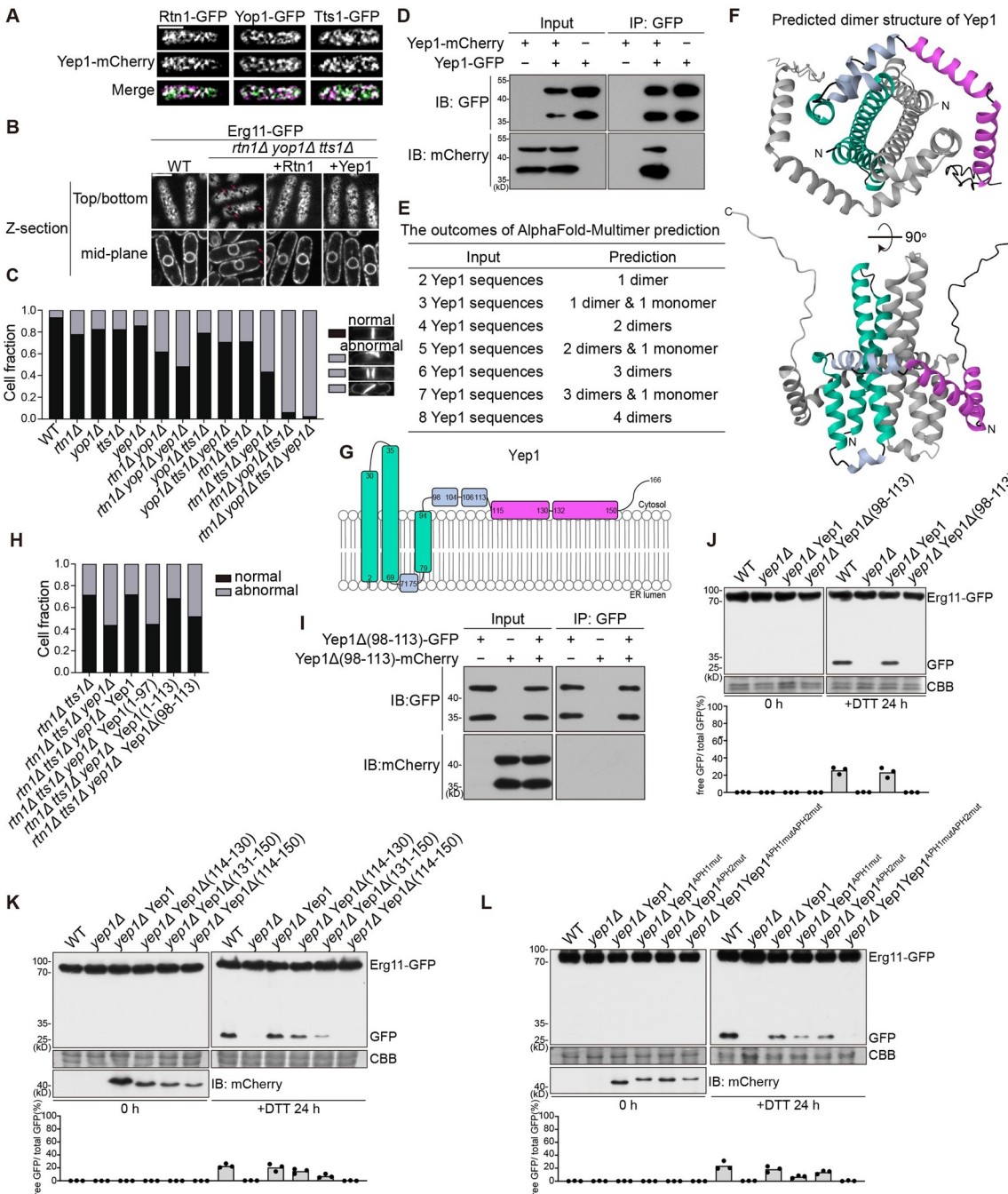

**Fig 3. The ER-phagy role of Yep1 relies on its abilities to self-interact and shape membranes and requires its C-terminal amphipathic helices.** (**A**) Yep1 colocalized with ER-shaping proteins. Single optical sections focused on the top (or bottom) of the cells are shown. Images were processed by deconvolution to allow better visualization of the cortical ER network. Bar, 5 μm. (**B**) The aberrant cortical ER morphology of *rtn1Δ yop1Δ tts1Δ* cells can be partially rescued by the overexpression of Yep1. A peripheral plane and a central plane of the same cells are shown. Red arrows denote the large holes in images of the peripheral planes or the extended gaps in images of the central planes. Images were processed by deconvolution to allow better visualization of the cortical ER network. Bar, 5 μm. (**C**) Quantification of the septum abnormality phenotypes (more than 200 cells with septa were examined for each sample). The septum abnormality phenotypes include long-axis septum, multiple septa, and tilted septum. Septa were visualized by calcofluor staining. (**D**) Yep1 exhibited self-interaction. Yep1-GFP was immunoprecipitated and co-immunoprecipitation of Yep1-mCherry was analyzed by immunoblotting. (**E**) The outcomes of predicting the structures of Yep1 oligomers using AlphaFold-Multimer. (**F**) The AlphaFold-Multimer-predicted Yep1 homodimer structure shown in the cartoon representation. One subunit is colored grey. The other subunit is colored to match the diagram in (**G**), with the transmembrane helices (TMHs) colored green, the 2 C-terminal amphipathic helices (APHs) colored magenta, and other α-helices colored light blue. (**G**) Topology model of Yep1. The secondary

structure is based on the predicted 3D structure shown in (**F**). The topology is based on the TOPCONS membrane protein topology prediction shown in S5C Fig. TMHs are colored green. The 2 C-terminal APHs are colored magenta. Other α-helices are colored light blue. (**H**) Quantification of the septum abnormality phenotypes of *rtn1Δ tts1Δ* cells, *rtn1Δ tts1Δ yep1Δ* cells, and *rtn1Δ tts1Δ yep1Δ* cells expressing full-length Yep1, Yep1 (1–97), Yep1 (1–113), or Yep1Δ (98–113) (more than 200 cells with septa were examined for each sample). (**I**) Yep1Δ (98–113) did not exhibit self-interaction in a co-immunoprecipitation analysis. (**J**) Yep1Δ (98–113) was unable to rescue the ER-phagy defect of *yep1Δ*. The blot images are representatives of triplicate experiments. Quantitation of triplicate experiments is shown below the images. (**K**) Internal deletion of both but not either one of the 2 helices in Yep1 (114–150) abolished the ER-phagy function of Yep1. The blot images are representatives of triplicate experiments. Quantitation of triplicate experiments is shown below the images. Proteins expressed in *yep1Δ* were tagged with mCherry, and their expression levels were analyzed by immunoblotting using an antibody against mCherry. (**L**) The amphipathic nature of APHs is important for the ER-phagy function of Yep1. Yep1$^{APH1mut}$ harbors the I117D, F120D, and L124D mutations; Yep1$^{APH2mut}$ harbors the A137D, V141D, and L148D mutations; Yep1$^{APH1mutAPH2mut}$ harbors the I117D, F120D, L124D, A137D, V141D, and L148D mutations (see helical wheel representations in S6I Fig). The blot images are representatives of triplicate experiments. Quantitation of triplicate experiments is shown below the images. Proteins expressed in *yep1Δ* were tagged with mCherry, and their expression levels were analyzed by immunoblotting using an antibody against mCherry. Numerical data underlying panels C, H, and J-L can be found in S1 Data, and raw images for panels D and J-L can be found in S1 Raw Images.

In the result of the co-immunoprecipitation assay, both Yep1-GFP and Yep1-mCherry appeared as doublets, and the lower band of Yep1-mCherry was co-immunoprecipitated with Yep1-GFP (**Fig 3D**). Based on the apparent molecular weights, the upper band and the low band likely correspond to the full-length protein and an N-terminally cleaved protein, respectively. To assess where the cleavage occurred, we expressed 2 N-terminally truncated forms, Yep1 (35–166)-mCherry and Yep1 (79–166)-mCherry (**S5D Fig**). Yep1 (35–166)-mCherry appeared as a doublet, whereas Yep1 (79–166)-mCherry appeared as a single band running slightly lower than the lower doublet band (**S5E and S5F Fig**), indicating that the cleavage likely occurred shortly upstream of residue 79, between the second and third TMH (**S5D Fig**). Neither Yep1 (35–166)-mCherry nor Yep1 (79–166)-mCherry was co-immunoprecipitated with Yep1-GFP (**S5E and S5F Fig**), suggesting that, consistent with the predicted structure of the Yep1 dimer, the N-terminal region of Yep1 is necessary for self-interaction. We speculate that the cleavage, which possibly occurred during protein extraction, did not cause dissociation of the N-terminal region of Yep1 under the non-denaturing immunoprecipitation conditions and therefore did not affect self-interaction.

Also consistent with the predicted structure of the Yep1 dimer, Yep1 (1–97) failed to exhibit self-interaction in the co-immunoprecipitation analysis (**S5G Fig**), whereas Yep1(1–113) was able to self-interact (**S5H Fig**). The septum positioning defect of *rtn1Δ tts1Δ yep1Δ* was rescued to the level of *rtn1Δ tts1Δ* by the expression of Yep1(1–113), but not Yep1(1–97) (**Fig 3H**), suggesting that the membrane-shaping ability of Yep1 depends on its self-interaction. Internal deletion of residues 98–113 abolished the abilities of Yep1 to self-interact and to rescue the septum positioning defect of *rtn1Δ tts1Δ yep1Δ* (**Fig 3H and 3I**). Moreover, Yep1 lacking residues 98–113 can no longer support ER-phagy (**Fig 3J**). Together, these results suggest that Yep1 self-interaction is important for its membrane-shaping ability and imply that the membrane-shaping ability is important for its role in ER-phagy.

## Amphipathic helices of Yep1 are essential for its ER-phagy function

Even though Yep1 (1–113) can self-interact and possesses the membrane-shaping ability, it cannot support ER-phagy in *yep1Δ* cells (**S6A Fig**). In contrast, Yep1 (1–131) and Yep1(1–150), which contained 1 and 2 additional C-terminal long helices, respectively, can support ER-phagy. Internal deletion of both but not either one of these 2 helices abolished the ER-phagy function of Yep1, suggesting that these 2 long helices play redundant roles for the ER-phagy function of Yep1 (**Fig 3K**). Consistent with the results obtained on Yep1 (1–113), the internal

deletion mutant Yep1Δ(114–150), which lacks both long helices, and Yep1(1–150) can self-interact and can rescue *rtn1Δ tts1Δ yep1Δ* to the same extent as full-length Yep1 (**S6B–S6E Fig**).

HeliQuest analyses of the C-terminal helices and visual inspection of the AlphaFold-predicted structure indicated that these 2 long helices are amphipathic helices (APHs), whereas the 2 upstream short helices do not exhibit obvious amphipathicity [57] (**S6F–S6H Fig**). To examine whether the amphipathic nature of these 2 APHs is functionally important, we substituted 3 hydrophobic amino acids with aspartates in each APH to disrupt their hydrophobic face [58,59] (**S6I Fig**). Mutating the first APH substantially weakened, but did not abolish, the ER-phagy function of Yep1 (**Fig 3L**). Mutating the second APH slightly weakened the ER-phagy function, while mutating both APHs rendered Yep1 nonfunctional in ER-phagy (**Fig 3L**). Together, these results demonstrate that these APHs are redundantly essential for the ER-phagy function of Yep1.

## REEP1-4 subfamily proteins and Atg40 share the same ER-phagy function with Yep1

To understand whether the structural features of Yep1 are conserved in its homologs, we applied the same analyses, including the inspection of AlphaFold-predicted structures, TOP-CONS prediction of membrane topology, and HeliQuest analysis of APHs, to several other representative REEP family proteins (**Fig 4A and 4B**). Consistent with a previous report [53], our analyses showed that REEP1-4 subfamily proteins have 3 TMHs, whereas REEP5-6 subfamily proteins have 4 TMHs. They all contain APHs in the C-terminal cytoplasmic region. REEP5-6 subfamily proteins also possess APHs in the N-terminal cytoplasmic region.

In *S. pombe*, as our data show (**S4F Fig**), the REEP5-6 subfamily member Yop1 cannot substitute for the ER-phagy function of the REEP1-4 subfamily member Yep1. We examined whether heterologously expressing human REEP family proteins can suppress the ER-phagy defect of *yep1Δ*. Remarkably, REEP1 and REEP3 fully suppressed *yep1Δ*, and REEP2 and REEP4 exhibited partial suppression (**Fig 4C**). In contrast, REEP5 and REEP6 showed no suppression (**Fig 4C**). These results suggest that REEP1-4 subfamily proteins, but not REEP5-6 subfamily proteins, share a conserved ER-phagy function with *S. pombe* Yep1.

Even though *S. cerevisiae* does not have an obvious REEP1-4 subfamily member, a known ER-phagy receptor in *S. cerevisiae*, Atg40, resembles REEP1-4 subfamily proteins in the number and topology of its TMHs [29]. Our analysis showed that it also possesses APHs downstream of its 3 TMHs (**Figs 4A and 4B and S7A**). To understand the relationship between Atg40 and REEP1-4 subfamily proteins, we surveyed the species distribution of Atg40 homologs and REEP1-4 subfamily proteins in the *Ascomycota* phylum (**Fig 4D**). PSI–BLAST-detectable sequence homologs of Atg40 (hereafter referred to as Atg40 proteins) were only found in budding yeast species belonging to the *Saccharomycetaceae* family. Interestingly, these species all lack a REEP1-4 subfamily protein. In contrast, *Ascomycota* species outside of the *Saccharomycetaceae* family all have at least 1 REEP1-4 subfamily protein. Remarkably, within the subphylum *Saccharomycotina* (budding yeasts), species not belonging to the *Saccharomycetaceae* family all possess 1 REEP1-4 subfamily protein harboring a C-terminal AIM, resembling the situation in Atg40 (**Figs 4D and S7B**). Furthermore, genes encoding these AIM-containing REEP1-4 subfamily proteins share synteny with *Saccharomycetaceae* genes encoding Atg40 proteins (**Fig 4E**). These observations indicate that Atg40 proteins are divergent orthologs of REEP1-4 subfamily proteins. Supporting this idea, a phylogenetic analysis showed that Atg40 proteins and *Ascomycota* REEP1-4 subfamily proteins fall into the same clade, and *Ascomycota* REEP5-6 subfamily proteins fall into a sister clade (**S7C Fig**).

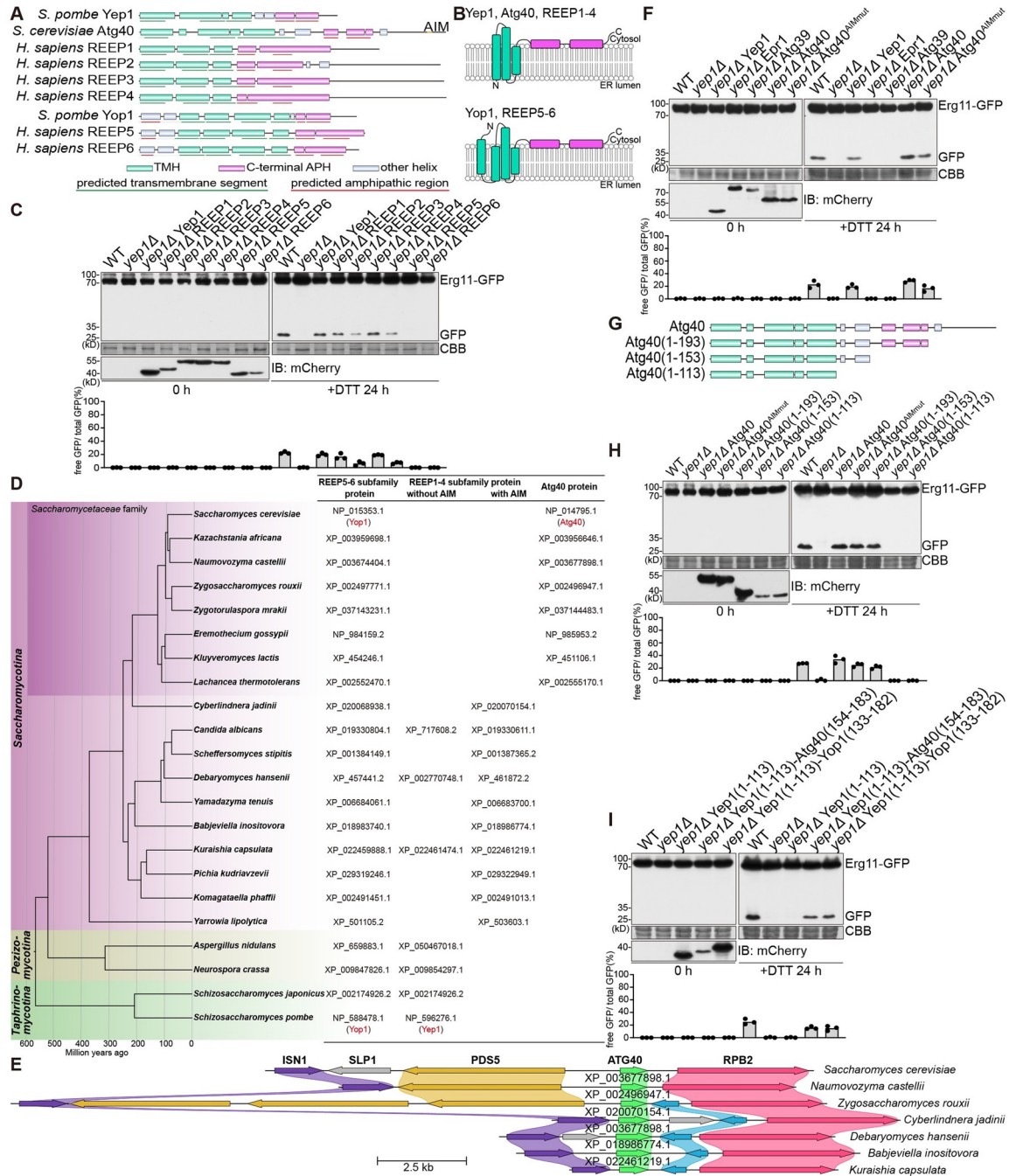

**Fig 4. REEP1-4 subfamily proteins and Atg40 share the same ER-phagy function with Yep1. (A)** Schematics of predicted structures of representative members of the REEP family and *S. cerevisiae* Atg40. The depicted structural elements are based on AlphaFold-predicted structures, TOPCONS prediction of membrane topology, and HeliQuest analysis of APHs. **(B)** Topology models of representative REEP family proteins and Atg40. Transmembrane helices (TMHs) are colored green. C-terminal APHs are colored magenta. **(C)** The ER-phagy defect of *yep1Δ* can be suppressed by expressing any one of the 4 human REEP1-4 subfamily proteins but not by expressing human REEP5 or REEP6. The blot images are representatives of triplicate experiments. Quantitation of triplicate experiments is shown below the images. Proteins expressed in *yep1Δ* were tagged with mCherry, and their expression levels were analyzed by immunoblotting using an antibody against mCherry. **(D)** *S. cerevisiae* Atg40 and its PSI–BLAST-detectable sequence homologs in other *Saccharomycetaceae* species (referred to as Atg40 proteins here) are likely divergent REEP1-4 subfamily proteins. Shown on the left is a time-calibrated species tree of representative *Ascomycota* species [60]. REEP5-6 subfamily proteins, REEP1-4 subfamily proteins, and Atg40 proteins present in these species are listed on the right. All Atg40 proteins and some REEP1-4 subfamily proteins harbor a C-terminal AIM (see S7B Fig). **(E)** *Saccharomycetaceae* genes encoding Atg40 proteins share synteny with non-*Saccharomycetaceae* genes encoding AIM-containing REEP1-4 subfamily proteins. Syntenic genes are shown as colored arrows, and

nonsyntenic genes are shown as grey arrows. The names of *Saccharomyces cerevisiae* genes are shown in bold. For the other species, the accession numbers of Atg40 proteins and AIM-containing REEP1-4 subfamily proteins are shown on top of the corresponding genes, which are denoted by green arrows. **(F)** Expressing *S. cerevisiae* Atg40 rescued the ER-phagy defect of *yep1Δ* in a manner independent of its Atg8-interacting motif (AIM). Atg40$^{AIMmut}$ harbors the Y242A and M245A mutations. The blot images are representatives of triplicate experiments. Quantitation of triplicate experiments is shown below the images. Proteins expressed in *yep1Δ* were tagged with mCherry, and their expression levels were analyzed by immunoblotting using an antibody against mCherry. **(G)** Schematics of wild-type and truncated Atg40. **(H)** Atg40 (1–193) but not Atg40 (1–153) rescued the ER-phagy defect of *yep1Δ*. The blot images are representatives of triplicate experiments. Quantitation of triplicate experiments is shown below the images. Proteins expressed in *yep1Δ* were tagged with mCherry, and their expression levels were analyzed by immunoblotting using an antibody against mCherry. **(I)** Fusing an APH-containing fragment from either *S. cerevisiae* Atg40 or *S. pombe* Yop1 to Yep1 (1–113) can partially restore its ER-phagy function. The blot images are representatives of triplicate experiments. Quantitation of triplicate experiments is shown below the images. Proteins expressed in *yep1Δ* were tagged with mCherry, and their expression levels were analyzed by immunoblotting using an antibody against mCherry. Numerical data underlying panels C, F, H, and I can be found in S1 Data, and raw images for panels C, F, H, and I can be found in S1 Raw Images.

Consistent with the results of the phylogenetic analysis, heterologous expression of Atg40 in *S. pombe* rescued the ER-phagy defect of *yep1Δ* (**Fig 4F**). In contrast, the *S. cerevisiae* nucleophagy receptor Atg39 failed to rescue *yep1Δ*. The ability of Atg40 to rescue *yep1Δ* is independent of its AIM (**Fig 4F**). Interestingly, heterologous expression of Atg40 in *S. pombe* can also rescue the ER-phagy defect of *epr1Δ*, and this rescue requires its AIM (**S7D Fig**). Moreover, Atg40 can even rescue the ER-phagy defect of the *yep1Δ epr1Δ* double mutant (**S7E Fig**), suggesting that Atg40 fulfills the combined roles of Yep1 and Epr1 in ER-phagy.

We examined the role of the APHs in Atg40 by truncation analysis. Atg40 (1–193), which lacks the C-terminal disordered tail, was still able to rescue the ER-phagy defect of *yep1Δ* (**Fig 4G and 4H**). In contrast, Atg40(1–153), which lacks the 2 C-terminal APHs, failed to support ER-phagy in *yep1Δ* cells (**Fig 4G and 4H**). To address the question why REEP1-4 subfamily proteins and Atg40, but not REEP5-6 subfamily proteins, can substitute for Yep1, we replaced the C-terminal region of Yep1 with a C-terminal APH-encompassing segment of *S. cerevisiae* Atg40 or *S. pombe* Yop1. The ER-phagy defect of *yep1Δ* can be suppressed to a similar extent by the 2 mosaic proteins (**Figs 4I and S7F**), indicating that the C-terminal APHs of REEP5-6 subfamily proteins are capable of supporting ER-phagy. One possibility is that the extra TMH and/or the cytoplasmic tail in the N-termini of REEP5-6 subfamily proteins are incompatible with ER-phagy. Supporting this possibility, we found that adding the N-terminal region of Yop1 upstream of its second TMH or only the first TMH of Yop1 to the N-terminus of Yep1 disrupted the ER-phagy function of Yep1 (**S7F and S7G Fig**). However, the presence of the extra N-terminal sequence may not be the only reason why REEP5-6 subfamily proteins cannot support ER-phagy, as removing the N-terminal region of Yop1 upstream of its second TMH did not render it capable of substituting for Yep1 (**S7F and S7H Fig**).

## Discussion

In this study, through an imaging-based chemical mutagenesis screen, we identify Yep1 as an essential factor for ER-phagy. Our follow-up investigation reveals that Yep1 is crucial for ER-phagy/nucleophagy under ER-stress and starvation conditions but is dispensable for bulk autophagy. In the absence of Yep1, the recruitment of the autophagic machinery at the early phase of ER-phagy/nucleophagy occurs normally, but ER-phagy/nucleophagy cargo structures without surrounding autophagic membranes accumulate in the cytoplasm, indicating that Yep1 plays a critical role in the autophagosomal enclosure of cargos during ER-phagy/nucleophagy.

The formation of nucleophagy cargo structures that accumulate in *yep1Δ* cells requires Atg5. This is possibly because the budding of the cargo structures depends on the local assembly of the isolation membrane by the Atg machinery. In *yep1Δ* cells, the outer membranes, but

not the inner membranes of the nucleophagy cargo structures, remain continuous with the nuclear envelope/ER network. We propose 2 alternative hypotheses to explain this (**S8 Fig**). The first posits that in *yep1Δ* cells, fission of the inner membranes, but not the outer membranes, occurs at the neck of the budded cargo structures. This results in the formation of luminal vesicles. These vesicles may move along cytoplasmic ER tubules or be pushed away as ER tubules form and extend on the source compartment side. The second hypothesis proposes that the cargo structures fully separate from the source compartments initially before reassociating with the ER network through homotypic fusion. The first hypothesis more readily explains the autophagosomal enclosure defect.

The best understood ER-phagy factors are ER-phagy receptors, which mediate the recruitment of the autophagic machinery. Several ER-phagy receptors have been shown to have additional functions. For example, FAM134 family proteins and RTN3L promote the remodeling and fragmentation of ER membranes [15,16,18,27,28,30,31], and Atg40 bends ER membranes to facilitate ER packaging into autophagosomes [29]. Non-receptor ER-phagy factors can be classified into 2 types based on their functions. The first type promotes ER-phagy through regulating ER-phagy receptors. Examples of this type include Ire1 that up-regulates the protein level of Epr1 [26], CK2 that enhances the Atg8-binding affinity of TEX264 [61], and CAMK2B that promotes FAM134B oligomerization [28]. The second type, mainly studied in *S. cerevisiae*, acts in concert with ER-phagy receptors to promote the formation of ER-phagy sites or the packaging of ER membranes into autophagosomes. This type of factors includes Lnp1 [62], the Lst1-Sec23 complex [63], Vps13 [32], and a group of proteins regulating actin assembly at sites of contact between the cortical ER and endocytic pits [64]. The loss of any essential ER-phagy factors invariably results in the failure to form autophagosomes containing ER membranes. However, it has not been reported before that ER-phagy cargo structures without surrounding autophagic membranes accumulate in an ER-phagy mutant. Thus, our findings reveal an unexpected mechanism ensuring the successful autophagosomal enclosure of cargos during ER-phagy.

Compared to ER-phagy, nucleophagy is more poorly understood. It is unclear to what extent nucleophagy shares the same underlying mechanisms with ER-phagy. *S. cerevisiae* Atg39 is the only known autophagy receptor with a dedicated role in nucleophagy [12]. Atg39 has non-receptor functions in linking the inner and outer nuclear membranes and in deforming the nuclear envelope [13,65]. No non-receptor factors required for nucleophagy have been reported. Our findings here show that in *S. pombe*, Epr1 serves as an autophagy receptor for both nucleophagy and ER-phagy, and Yep1 is required for autophagosomal enclosure of cargos during both nucleophagy and ER-phagy, suggesting that these 2 processes share a common set of factors.

The exact role of Yep1 in ER-phagy/nucleophagy remains unclear. Here, we discuss 2 possibilities based on the requirement of its membrane-shaping ability: Yep1 may remodel cargo membranes, or it may help shaping autophagic membranes. These 2 possibilities are not mutually exclusive.

In the first possibility, as an integral ER membrane protein, Yep1 may exert its function on the ER/nuclear envelope. This proposed role of Yep1 is similar to the non-receptor roles proposed for RTN3L and FAM134B in mammalian cells and Atg40 in *S. cerevisiae*. The fact that the outer membranes of the cargo structures accumulating in *yep1Δ* cells remain continuous with the nuclear envelope/ER network suggests that Yep1 may promote the fission of the outer membranes of cargo structures. Supporting a role of Yep1 on the ER membrane, in an immunoprecipitation coupled with mass spectrometry analysis using Yep1-GFP as bait, we found that a large number of ER membrane proteins were co-immunoprecipitated with Yep1 (**S1 Table**). Among them are Scs2 and Scs22, 2 VAP-family proteins responsible for the ER

localization of the ER-phagy receptor Epr1 [26]. It is possible that the VAP proteins connect Yep1 to Epr1. Additionally, several ER-shaping proteins, including Rtn1, Yop1, Tts1, Sey1 (atlastin homolog), and Lnp1 (lunapark homolog), co-immunoprecipitated with Yep1. This is analogous to the situation in mammalian cells where FAM134B clusters with other ER-shaping proteins to promote ER-phagy [30,31]. These potential Yep1 interactions warrant further investigation in the future.

In the second possibility, some Yep1 molecules may relocalize from the ER to the isolation membrane and play a role in shaping the isolation membrane. We speculate that this relocalization may take the route of COPII vesicles, which have been shown to transport an integral ER membrane protein to the autophagic membranes [66]. If Yep1 functions on the isolation membrane, it begs the question: Why is Yep1 essential for ER-phagy/nucleophagy but dispensable for bulk autophagy? We speculate that one possible explanation is that different types of autophagy may utilize different membrane sources for the isolation membrane. As a result, the isolation membrane for ER-phagy/nucleophagy may have a protein composition different from the isolation membrane for bulk autophagy, and Yep1 is not important for bulk autophagy because there are other factors playing a similar function on the isolation membrane for bulk autophagy. Another possibility is that the size and shape of ER-phagy/nucleophagy cargos impose a special requirement for the shape of the isolation membrane, and Yep1 is needed to meet this requirement.

During the preparation and submission of this manuscript, Wang and colleagues and Fukuda and colleagues reported the identification of Yep1 as an autophagy factor [67,68]. Wang and colleagues show that Yep1 (called Rop1 in their paper) localizes to the isolation membrane, supporting the second possibility discussed above. There are several discrepancies between our study and that of Wang and colleagues'. Yep1 is found to be largely dispensable for bulk autophagy in our study but shown to be important for bulk autophagy by Wang and colleagues. In addition, we show that Atg40 can substitute for the role of Yep1 in ER-phagy, while Wang and colleagues show that Atg40 fails to suppress the sensitivity of *yep1Δ* to ER stress. The exact reasons of these discrepancies are unclear but may be related to differences in strain backgrounds and assaying conditions. Similar to our findings, Fukuda and colleagues showed that Yep1 (called Hva22 in their paper) is essential for ER-phagy but dispensable for bulk autophagy, and the ER-phagy function of Yep1 can be substituted by budding yeast Atg40. Neither Wang and colleagues nor Fukuda and colleagues reported the cargo structure accumulation phenotype that we observed in *yep1Δ* cells, likely because they did not examine the localization of nuclear proteins, which provides the clearest evidence of this phenotype.

Based on the phylogenetic analysis results shown here (**1B Fig**) and elsewhere [37–40], both the REEP1-4 subfamily and the REEP5-6 subfamily exist in the common ancestor of animals and fungi. The findings that human REEP1-4, but not REEP5-6, can substitute for the ER-phagy function of Yep1 indicate that the REEP1-4 subfamily may have an ancestral role in ER-phagy. Our analyses of the evolutionary relationships of REEP family proteins in the *Ascomycota* phylum (**Figs 4D and 4E and S7B and S7C**) suggest that in the *Saccharomycotina* subphylum (budding yeasts), the ER-phagy role of the REEP1-4 subfamily proteins is further enhanced by the acquisition of a C-terminal AIM so that they can also act as ER-phagy receptors. For reasons unclear, substantial sequence divergence happened to the REEP1-4 subfamily proteins in the common ancestor of the *Saccharomycetaceae* family, giving rise to the Atg40 proteins. Further studies will be needed to understand to what extent REEP1-4 subfamily proteins in different species share common ER-phagy functions and mechanisms.

## Methods

### Strain and plasmid construction

Fission yeast strains used in this study are listed in S2 Table, and plasmids used in this study are listed in S3 Table. The genetic methods for strain construction and composition of media are as previously described [69]. Deletion strains were generated by PCR-based gene targeting. The strain containing 1 additional copy of each of 23 core autophagy genes (*atg1*, *atg2*, *atg3*, *atg4*, *atg5*, *atg6*, *atg7*, *atg8*, *atg9*, *atg10*, *atg11*, *atg12*, *atg13*, *atg14*, *atg16*, *atg17*, *atg18a*, *atg18b*, *atg18c*, *atg101*, *vps34*, *vps15*, and *ctl1*) was constructed using the CRISPR-Cas9 system [70]. The plasmids expressing proteins fused with different N-terminal or C-terminal tags (GFP, CFP, mECtrine, mCherry, mTurquoise2) under exogenous promoters (*P41nmt1*, *Pnmt1*, or *Padh1*) were constructed utilizing modified pDUAL vectors [71] or modified SIV vectors [72]. The plasmids expressing Erg11-AIM$^{art}$ were based on modified SIV vectors. AIM$^{art}$ corresponds to 3×EEEWEEL [73].

### Screening for mutants defective in ER-phagy

Log phase cells were harvested and resuspended in 0.3 ml of TM buffer (50 mM Tris-maleate). Mutagenesis was induced by adding 0.1 ml of 2 mg/ml N-methyl-N′-nitro-N-nitrosoguanidine (MNNG) to the cell suspension and incubating at room temperature for 60 minutes. The mutagenized cells were then plated on the EMM medium and incubated at 30°C for 4 days. Small and medium-sized colonies were selected, transferred to 96-well deep-well plates, grown for 24 hours in EMM liquid medium at 30°C, and treated with 10 mM DTT for 12 hours to induce ER-phagy. A high-content imaging system (Opera LX, PerkinElmer) was used for observing the subcellular localization of Ost4-GFP. Candidate mutants defective in vacuolar relocalization of Ost4-GFP were further evaluated using a DeltaVision PersonalDV system (Applied Precision). Strains whose phenotypes were confirmed by reexamination were backcrossed, and the backcrossed progeny were analyzed via next-generation sequencing-assisted bulk segregant analysis [74]. For each backcross, mutations enriched in ER-phagy defective progeny were considered as the candidate phenotype-causing mutations.

### Fluorescence microscopy

Live-cell imaging was performed using a DeltaVision PersonalDV system (Applied Precision) and a Dragonfly high-speed confocal microscope system (Andor Technology). The DeltaVision system was equipped with a 100×, 1.4-NA objective, an mCherry/YFP/CFP filter set, and a Photometrics EMCCD camera. The Dragonfly system was equipped with a 100×, 1.4-NA objective, an mCherry/YFP/CFP filter set, an mCherry/GFP filter set, and a Sona sCMOS camera. Image analysis was conducted using the SoftWoRx software and Fiji (ImageJ).

### Immunoblotting-based protein processing assay

Approximately 5.0 OD$_{600}$ units of cells expressing a GFP-tagged protein (Erg11, Ish1, or Rtn1), an mECtirine-tagged protein (Pus1), or a CFP-tagged protein (Atg8 or Tdh1) were harvested. The cells were mixed with 300 μl of 20% trichloroacetic acid (TCA) and lysed by beating with 0.5-mm-diameter glass beads using a FastPrep instrument at a speed of 6.5 m/s for 3 cycles of 20 seconds each. The cell lysate was centrifuged and the pellet was resuspended in HU buffer (150 mM Tris–HCl, 6% SDS, 6 M urea, 10% 2-mercaptoethanol, 0.002% bromophenol blue (pH 6.8)) and incubated at 42°C for 20 minutes. The samples were then separated by 10% SDS-PAGE and immunoblotted with antibodies. The antibodies used for immunoblotting were

anti-GFP mouse monoclonal antibody (1:3,000 dilution, Roche, 11814460001) and anti-mCherry mouse monoclonal antibody (1:3,000 dilution, Huaxingbio, HX1810).

## Electron microscopy

For regular TEM analysis, 50 $OD_{600}$ units of cells were harvested after being starved for 12 hours or treated with 10 mM DTT for 12 hours. Cells were fixed with 1% glutaraldehyde and 4% KMnO4 and dehydrated through a graded ethanol series. The samples were then embedded in Spurr's resin [75]. Thin sections were examined using an FEI Tecnai G2 Spirit electron microscope equipped with a Gatan 895 4k×4k CCD camera. The diameters of the ring-shaped structures were determined using the method previously used for measuring the sizes of autophagosomes or autophagic bodies [43,76]. $P$ values were calculated using Welch's $t$ test. For electron microscopy analysis employing the genetically encoded EM tag MTn, samples were prepared as previously described [46]. Briefly, 20 OD600 units of cells expressing Ost4-MTn were harvested after treatment with 10 mM DTT for 12 hours. Cells were incubated with 3 mM dithiodipropionic acid (DTDPA) in 0.1 M PIPES at 4°C for 30 minutes and then treated with 0.5 mg/ml zymolyase 20T and 50 mM DTT in PBS buffer for 10 minutes to remove the cell wall. The zymolyase-treated cells were permeabilized with 0.05% Triton X-100 for 5 minutes and processed for gold nanoparticle synthesis. Cells were mixed with 60 mM 2-mercaptoethanol, 0.5 mM $HAuCl_4$, 50 mM diphenylethylenediamine (DPEN), and 10 μM $NaBH_4$ and subjected to standard high-pressure freezing/freeze-substitution fixation. After resin infiltration, embedding, polymerization, and thin sectioning, the samples were used for EM imaging as described above.

## Pil1 co-tethering assay

To investigate the Atg8-interacting ability, Yep1 and Epr1 were fused to GFP as the prey, and Atg8 was fused to Pil1-mCherry as the bait [45]. Log-phase cells coexpressing both proteins were then analyzed by fluorescence microscopy.

## Protein depletion using the auxin-inducible degron (AID)

Cells cultured in EMM liquid media at 30°C were used. Prior to observation, cells in EMM liquid media were treated with 1 μM 5-adamantyl-IAA for 1.5 hours at 30°C [48].

## Fluorescence recovery after photobleaching (FRAP)

FRAP experiments were performed using a LSM800 confocal microscope system (Carl Zeiss) equipped with a 63× oil objective. Regions of 1 μm × 1 μm were photobleached for 20 iterations by a 561-nm laser at 100% output intensity. After photobleaching, the samples were imaged every 4.4 seconds for 120 seconds. For quantification, the fluorescence intensity before photobleaching was set to 100%. The background fluorescence was subtracted. The fluorescence decay during imaging was compensated by calculating the fluorescence decay ratio of unbleached regions and applying the ratio as a normalization factor.

## Immunoprecipitation

Approximately 100 $OD_{600}$ units of log-phase cells were harvested. Cells were mixed with 150 μl of lysis buffer (50 mM HEPES–NaOH (pH 7.5), 150 mM NaCl, 1 mM EDTA, 1 mM DTT, 1 mM PMSF, 0.05% NP-40, 10% glycerol, 1×Roche protease inhibitor cocktail) and were lysed by beating with 0.5-mm-diameter glass beads using a FastPrep instrument at a speed of 6.5 m/s for 3 cycles of 20 seconds each. The lysate was incubated with GFP-Trap agarose beads

at 4˚C for 3 hours. The beads were washed twice and proteins were eluted by boiling in SDS-PAGE loading buffer. Samples were separated by 10% SDS-PAGE and analyzed by immunoblotting using anti-GFP and anti-mCherry antibodies described above. For the immunoprecipitation coupled with mass spectrometry (IP-MS) experiment using Yep1-GFP as bait, immunoprecipitated samples were processed and mass spectrometry was performed as described previously [26]. S1 Table lists ER-localizing proteins (GO:0005783) with a spectral count of at least 30 in the Yep1-GFP IP sample and more than 6-fold higher than the spectral count in a control IP sample.

### Bifluorescence complementation (BiFC) assays

To investigate the self-interaction of Yep1, the N-terminal Venus fragment (VN173) and the C-terminal Venus fragment (VC155) were fused to Yep1, respectively [77]. Log-phase cells coexpressing the VN173 fusing protein and the VC155 fusing protein were analyzed by fluorescence microscopy.

### Phylogenetic analysis and synteny analysis

Protein sequences were aligned using the L-INS-i iterative refinement algorithm of MAFFT on the online MAFFT server (https://mafft.cbrc.jp/alignment/server/) [78]. The maximum likelihood trees were calculated using IQ-TREE (version 2.1.3) for Mac OS X [79]. Trees were rooted by midpoint rooting and visualized using FigTree (version 1.4.4) (http://tree.bio.ed.ac.uk/software/figtree/). Synteny plot was generated using clinker on the CAGECAT webserver (https://cagecat.bioinformatics.nl/tools/clinker) [80].

### Prediction of protein structures, transmembrane topology, and amphipathic helices

The structures of the Yep1 monomer and oligomers were predicted using AlphaFold-Multimer (version 2.2.0) with default parameters [55]. The structure with the highest confidence score among the predicted output was selected for further analysis. The predicted structures were visualized using the Mol* Viewer (version 2.5.0). The transmembrane topology was predicted using the TOPCONS web server (https://topcons.cbr.su.se/pred/) [56]. The amphipathic nature of helices was analyzed using the HeliQuest web server (https://heliquest.ipmc.cnrs.fr/cgi-bin/ComputParams.py) [57].

### Supporting information

**S1 Fig. The identification and initial characterization of Yep1 as a factor required for ER-phagy and nucleophagy. (A)** Bulk segregant analysis identifying a mutation in *SPBC30D10.09c* (*yep1*) as a candidate phenotype-causing mutation in an ER-phagy defective mutant. The scatter plot depicts the reference allele frequencies at SNP sites in the pool of the ER-phagy defective segregants derived from a cross between the mutant strain and a wild-type strain. The T17M mutation in *SPBC30D10.09c* (*yep1*) is highlighted in red. **(B)** Subcellular localization of Yep1-mCherry expressed from the *P41nmt1* promoter. Log-phase *yep1Δ* cells coexpressing Yep1-mCherry and the ER marker Erg11-GFP were examined by fluorescence microscopy. Bar, 5 μm. **(C)** Yep1-mECitrine formed puncta colocalizing with Epr1 and Atg8 double positive puncta after ER-phagy induction by nitrogen starvation and DTT treatment. Red arrows denote puncta where Yep1-mECitrine, Epr1-mCherry, and mTurquoise2-Atg8 colocalize. Bar, 5 μm. **(D)** Quantification of the percentage of Epr1 and Atg8 double positive (Atg8+/Epr1+) puncta that are also positive for Yep1 in the analysis shown in (**C**) (more than

100 Atg8+/Epr1+ puncta were examined for each sample). **(E)** Autophagic processing of the bulk autophagy marker Tdh1-CFP was largely normal in *yep1Δ* cells. **(F)** Electron microscopy analysis of starved and DTT-treated *fsc1Δ* and *fsc1Δ yep1Δ* cells. N, nucleus; V, vacuole; A, autophagosome. Double-ring structures are denoted by pink arrows. Bar, 1 μm. **(G)** Quantification of the number of double-ring structures per cell in the analysis shown in **(E)** (more than 50 cells with autophagosomes were examined for each sample). **(H)** Yep1 did not interact with Atg8 in a Pil1 co-tethering assay. Log-phase cells coexpressing the bait (Pil1-mCherry or Pil1-mCherry-Atg8) and the prey Yep1-GFP were examined by fluorescence microscopy. Cells coexpressing Pil1-mCherry-Atg8 and Epr1-GFP served as a positive control. Peripheral planes of the cells were imaged. Bar, 5 μm. **(I)** Yep1 is not required for the DTT-induced increase of the protein level of Epr1. *isp6Δ psp3Δ* background, which lacks vacuolar protease activities, was used to prevent the degradation of Epr1. Endogenously tagged Epr1-mCherry was analyzed by immunoblotting. **(J)** Yep1 did not interact with Epr1 in a Y2H assay. Crb2 served as a specificity control, and the self-interaction of Crb2 and the interaction between Epr1 and Atg8 served as positive controls. **(K)** Ectopic expression of Erg11-AIM$^{art}$ but not Yep1 from the *P41nmt1* promoter suppressed the ER-phagy defect of *epr1Δ*. **(L)** Ectopic expression of Erg11-AIM$^{art}$ or Epr1 did not rescue the ER-phagy defect of *yep1Δ*. Numerical data underlying panels A, D, and G can be found in S1 Data, and raw images for panels E, I, K, and L can be found in S1 Raw Images.
(TIF)

**S2 Fig. ER-phagy and nucleophagy cargos accumulated in the cytoplasm of *yep1Δ* cells. (A)** Yep1 is not required for the colocalization of Epr1 and Atg8 at punctate structures shortly after ER-phagy induction. Wild-type and *yep1Δ* cells coexpressing Epr1-mCherry and mTurquoise2-Atg8 were examined by microscopy after 2.5-hour DTT or 1.5-hour starvation treatment. Bar, 5 μm. **(B)** Quantification of the colocalization between Bqt4 puncta and Pus1 puncta in the analysis shown in Fig 2C (more than 250 puncta were examined for each sample). **(C, D)** The vast majority of cytoplasmic Pus1 puncta **(C)** and Bqt4 puncta **(D)** in *yep1Δ* cells did not colocalize with Atg8 puncta. Bar, 5 μm. Over 200 Pus1 or Bqt4 puncta were examined per sample. **(E)** Electron microscopy analysis of nitrogen-starved and DTT-treated wild-type and *yep1Δ atg5Δ* cells. N, nucleus; V, vacuole; M, mitochondrion; A, autophagosome. Bar, 1 μm. **(F)** Quantification of the number of cytoplasmic ring-shaped structures in the analysis shown in Figs 2D and S2E (more than 30 cells were examined for each sample). Autophagosomes, which are ring-shaped structures juxtaposed to vacuoles, were excluded from this quantification. **(G)** Quantification of the diameters of the ring-shaped structures in *yep1Δ* cells in the analysis shown in Fig 2D and the inner rings in the double-ring structures in *fsc1Δ* cells in the analysis shown in S1F Fig. *P* values were calculated using Welch's *t* test. Numerical data underlying panels B-D, F, and G can be found in S1 Data.
(TIF)

**S3 Fig. ER-phagy and nucleophagy cargos not enclosed within autophagosomes accumulated in the cytosplasm of *yep1Δ* cells. (A)** Applying the degron protection assay on the cytosolic protein Pyk1-AID-mECitrine. Prior to observation, cells were treated with (+Ad-IAA) or without (−Ad-IAA) 5-adamantyl-IAA for 1.5 hours. BF, brightfield. Cpy1-mCherry is a vacuole lumen marker. Bar, 5 μm. **(B)** Applying the degron protection assay on Epr1-AID-mECitrine. BF, brightfield. Bar, 5 μm. **(C)** Applying the degron protection assay on Rtn1-AID-mECitrine. BF, brightfield. Bar, 5 μm. **(D)** Electron microscopy images of ER-phagy/nucleophagy cargo structures with filamentous membrane protrusions in *yep1Δ* cells. N, nucleus; V, vacuole. Pink arrows denote filamentous membrane protrusions that extend from the ER-

phagy/nucleophagy cargo structures. Bar, 1 μm.
(TIF)

**S4 Fig. Yep1 possesses the ability to shape the ER membrane. (A)** Colocalization between Yep1-mCherry and Rtn1-GFP, Yop1-GFP, or Tts1-GFP was quantitated using Pearson's correlation coefficient (PCC). The PCC values are presented as mean ± SD ($n$ = 10 cells). **(B)** Quantification of the percentages of cells with extended gaps in images of the midplane in the analysis shown in Fig 3B (more than 300 cells were examined for each sample). **(C, D)** Quantification of the septum abnormality phenotypes (more than 200 cells with septa were examined for each sample). **(E)** ER-phagy induced by DTT treatment or nitrogen starvation was largely normal in the absence of Rtn1, Yop1, and Tts1. **(F)** The ER-phagy defect of *yep1Δ* cells was not suppressed by the ectopic expression of Rtn1, Yop1, or Tts1. Proteins expressed in *yep1Δ* were tagged with mCherry, and their expression levels were analyzed by immunoblotting using an antibody against mCherry. Numerical data underlying panels A–D can be found in S1 Data, and raw images for panels E and F can be found in S1 Raw Images.
(TIF)

**S5 Fig. Yep1 self-interaction, the predicted structure of the Yep1 dimer, and the predicted topology of Yep1. (A)** Yep1 exhibited self-interaction in the BiFC assay. Log-phase cells expressing Yep1-VenusN173 alone, Yep1-VenusC155 alone, or both were examined by fluorescence microscopy. Bar, 5 μm. **(B)** The predicted aligned error (PAE) plot and pLDDT plot of the AlphaFold-Multimer-predicted structure of the Yep1 homodimer shown in Fig 3F. **(C)** TOPCONS membrane protein topology prediction for Yep1. **(D)** Schematics of wild-type and truncated Yep1. **(E)** Yep1 (35–166) did not interact with full-length Yep1 in a co-immunoprecipitation analysis. **(F)** Yep1 (79–166) did not interact with full-length Yep1 in a co-immunoprecipitation analysis. **(G)** Yep1 (1–97) did not exhibit self-interaction in a co-immunoprecipitation analysis. **(H)** Yep1 (1–113) exhibited self-interaction in a co-immunoprecipitation analysis. Numerical data underlying panels B and C can be found in S1 Data, and raw images for panel E-H can be found in S1 Raw Images.
(TIF)

**S6 Fig. APHs of Yep1 are essential for its ER-phagy function. (A)** Yep1 (1–131) or Yep1 (1–150), but not Yep1(1–113), is able to support ER-phagy. **(B)** Yep1 (1–150) exhibited self-interaction in a co-immunoprecipitation analysis. **(C)** Yep1Δ (114–150) exhibited self-interaction in a co-immunoprecipitation analysis. **(D)** Quantification of the septum abnormality phenotypes in *rtn1Δ tts1Δ* cells, *rtn1Δ tts1Δ yep1Δ* cells, and *rtn1Δ tts1Δ yep1Δ* cells expressing full-length Yep1, Yep1 (1–150), or Yep1Δ (114–150) (more than 200 cells with septa were examined for each sample). **(E)** Summary of the truncation and internal deletion analysis of Yep1. **(F)** Helical wheel representations of 2 APHs of Yep1. The helical wheels were generated using HeliQuest. Hydrophobic residues are colored in yellow, hydrophilic residues in blue (R and K), red (D and E), purple (T and S), and pink (N and Q), alanine in grey, and proline in green. The HeliQuest-calculated hydrophobic moment (μH) of the helix is shown. **(G)** The amphipathic nature of APHs is visualized in the AlphaFold-Multimer-predicted structure. The 2 APHs and the intervening amino acid are shown in the surface representation and are colored base on hydrophobicity. The rest of Yep1 is shown in the cartoon representation. **(H)** Helical wheel representation and the hydrophobic moment (μH) of residues 97–113 of Yep1. **(I)** Helical wheel representations and the hydrophobic moments (μH) of mutated APHs. Numerical data underlying panel D can be found in S1 Data, and raw images for panels A-C can be found in S1 Raw Images.
(TIF)

**S7 Fig. REEP1-4 subfamily proteins and Atg40 share the same ER-phagy function with Yep1. (A)** Helical wheel representations and the hydrophobic moments (µH) of the 2 C-terminal APHs in Atg40. **(B)** The alignment of the C-terminal AIM in the proteins whose names are colored red in (**C**). The AIM core motif is highlighted. **(C)** Phylogenetic relationships of REEP family proteins and Atg40 proteins in representative *Ascomycota* species. The sequences of REEP family proteins were retrieved by PSI–BLAST from the NCBI refseq_protein database using the sequences of *Yarrowia lipolytica* orthologs of *S. pombe* Yop1 and Yep1 as queries. The sequences of Atg40 proteins were retrieved by PSI–BLAST from the NCBI refseq_protein database using the sequence of *S. cerevisiae* Atg40 as query. A sequence alignment was generated using MAFFT, and a maximum likelihood tree was constructed using IQ-TREE. The tree was rooted using the REEP5-6 subfamily proteins as outgroup. Branch labels are the SH-aLRT support values (%) and the UFBoot support values (%) calculated by IQ-TREE. The names of proteins containing a C-terminal AIM are colored red. The scale bar indicates 0.8 substitutions per site. **(D)** *S. cerevisiae* Atg40 rescued the ER-phagy defect of *epr1Δ* in a manner dependent on its Atg8-interacting motif (AIM). Atg40^AIMmut harbors the Y242A and M245A mutations. Proteins expressed in *yep1Δ* were tagged with mCherry, and their expression levels were analyzed by immunoblotting using an antibody against mCherry. **(E)** *S. cerevisiae* Atg40 rescued the ER-phagy defect of *epr1Δ yep1Δ* in a manner dependent on its AIM. Proteins expressed in *yep1Δ* were tagged with mCherry, and their expression levels were analyzed by immunoblotting using an antibody against mCherry. **(F)** Schematics of wild-type and truncated Yop1. Yop1 (133–182) appears in Fig 4I. Yop1 [1–52] and Yop1 [35–52] appear in (**G**). Yop1 (53–182) appears in (**H**). **(G)** Adding an extra N-terminal transmembrane helix (TMH) to Yep1 disrupted its ER-phagy function. Yop1 [1–52] includes the N-terminal cytosolic region and the first TMH of Yop1. Yop1 [35–52] includes only the first TMH of Yop1. Proteins expressed in *yep1Δ* were tagged with mCherry, and their expression levels were analyzed by immunoblotting using an antibody against mCherry. **(H)** Removing the N-terminal region of Yop1 upstream of its second TMH did not render it capable of substituting for Yep1. Yop1 (53–182) lacks the N-terminal cytosolic region and the first TMH. Proteins expressed in *yep1Δ* were tagged with mCherry, and their expression levels were analyzed by immunoblotting using an antibody against mCherry. Raw images for panels D, E, G, and H can be found in S1 Raw Images.
(TIF)

**S8 Fig. Schematic depicting 2 hypotheses explaining the cargo structure accumulation phenotype of *yep1Δ*.** In wild-type cells, ER-phagy/nucleophagy cargos are sequestered into autophagosomes after their separation from the source compartments and are delivered to the vacuole through autophagosome–vacuole fusion. In the absence of Yep1, the recruitment of the autophagic machinery at the early phase of ER-phagy/nucleophagy occurs normally, but ER-phagy/nucleophagy cargos fail to be delivered to the vacuole. Instead, ER-phagy/nucleophagy cargo structures not enclosed within autophagosomes accumulate in the cytoplasm. The outer membranes of these structures remain continuous with the nuclear envelope-ER network. In hypothesis 1, we propose that fission of the outer membranes of ER-phagy/nucleophagy cargos fails to occur during cargo separation, resulting in the formation of luminal vesicles. These vesicles may move along the cytoplasmic ER tubules. In hypothesis 2, cargo separation happens but autophagosome enclosure somehow fails. Fully separated cargos reassociate with the ER network through homotypic fusion.
(TIF)

**S1 Table. The data of the IP-MS experiment using Yep1-GFP as bait.**
(XLSX)

**S2 Table. *S. pombe* strains used in this study.**
(XLSX)

**S3 Table. Plasmids used in this study.**
(XLSX)

**S1 Data. An Excel spreadsheet containing, in separate tabs, the numerical data underlying Figs** 1D–1J, 2I, 2J, 3C, 3H, 3J-3L, 4C, 4F, 4H, 4I, S1A, S1D, S1G, S2B–S2D, S2F, S2G, S4A–S4D, S5B, S5C, **and** S6D.
(XLSX)

**S1 Raw Images. Raw images supporting blot and gel results in Figs** 1D–1J, 3D, 3I-3L, 4C, 4F, 4H, 4I, S1E, S1I, S1K, S1L, S4E, S4F, S5E–S5H, S6A–S6C, S7D, S7E, S7G, **and** S7H.
(PDF)

# Acknowledgments

We are grateful to Yin Liu and He-Xia Luo of the NIBS electron microscopy facility for helps in EM analysis.

# Author Contributions

**Conceptualization:** Chen-Xi Zou, Li-Lin Du.

**Investigation:** Chen-Xi Zou, Zhu-Hui Ma, Zhao-Di Jiang, Zhao-Qian Pan, Dan-Dan Xu, Fang Suo, Guang-Can Shao, Meng-Qiu Dong, Li-Lin Du.

**Writing – original draft:** Chen-Xi Zou, Li-Lin Du.

**Writing – review & editing:** Chen-Xi Zou, Li-Lin Du.

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
