## [Editor Report · Decision Letter 0]

11 May 2023

Dear Dr Du, 

Thank you for submitting your manuscript entitled "Fission yeast ortholog of human REEP1-4 is required for autophagosomal enclosure of ER-phagy/nucleophagy cargos" for consideration as a Research Article by PLOS Biology. Please accept my apologies for the delay in getting back to you as we consulted with an academic editor about your submission.

Your manuscript has now been evaluated by the PLOS Biology editorial staff, as well as by an academic editor with relevant expertise, and I am writing to let you know that we would like to send your submission out for external peer review.

IMPORTANT: After our discussions with the Academic Editor, we would like to consider your manuscript as a Short Report at the journal. Upon resubmission (see details below), we ask that you please tick 'Short Report' as the article type and reduce the number of main figures to 4. This can be done by either combining some of the main figures or by moving some of the figures to the Supplementary. 

Before we can send your manuscript to reviewers, we need you to complete your submission by providing the metadata that is required for full assessment. To this end, please login to Editorial Manager where you will find the paper in the 'Submissions Needing Revisions' folder on your homepage. Please click 'Revise Submission' from the Action Links and complete all additional questions in the submission questionnaire.

Once your full submission is complete, your paper will undergo a series of checks in preparation for peer review. After your manuscript has passed the checks it will be sent out for review. To provide the metadata for your submission, please Login to Editorial Manager (https://www.editorialmanager.com/pbiology) within two working days, i.e. by May 13 2023 11:59PM.

Kind regards,

Richard

Richard Hodge, PhD

Associate Editor, PLOS Biology

rhodge@plos.org

PLOS

---

## [Decision Letter · Decision Letter 1]

15 Jun 2023

Dear Dr Du,

Thank you for your patience while your manuscript "Fission yeast ortholog of human REEP1-4 is required for autophagosomal enclosure of ER-phagy/nucleophagy cargos" was peer-reviewed at PLOS Biology. Please accept my sincere apologies for the delays that you have experienced during the peer review process. Your manuscript has now been evaluated by the PLOS Biology editors, an Academic Editor with relevant expertise, and by four independent reviewers. 

In light of the reviews, which you will find at the end of this email, we would like to invite you to revise the work to thoroughly address the reviewers' reports.

As you will see, the reviewers are generally positive about your study and think the findings are interesting and impactful for the field. Reviewer #3 suggests several experiments to strengthen the manuscript, such as analysing whether Yep1 accumulates at the site where the ER/nucleus vesicles form. Reviewer #4 raises concerns with the lack of quantification for the autophagy assays, as well as noting that the western blot data should be repeated in triplicate and quantified. Finally, Reviewer #2 raises several concerns with the overall strength of the mechanistic insights into the regulation of Yep1 function. However, after discussions with the academic editor, we will not make these mechanistic experiments essential to consider a revision, given the Short Report format of the manuscript. 

Given the extent of revision needed, we cannot make a decision about publication until we have seen the revised manuscript and your response to the reviewers' comments. Your revised manuscript is likely to be sent for further evaluation by all or a subset of the reviewers.

**IMPORTANT - SUBMITTING YOUR REVISION**

*Re-submission Checklist*

*Published Peer Review*

*PLOS Data Policy*

Please note that as a condition of publication PLOS' data policy (http://journals.plos.org/plosbiology/s/data-availability) requires that you make available all data used to draw the conclusions arrived at in your manuscript. If you have not already done so, you must include any data used in your manuscript either in appropriate repositories, within the body of the manuscript, or as supporting information (N.B. this includes any numerical values that were used to generate graphs, histograms etc.). For an example see here: http://www.plosbiology.org/article/info:doi%2F10.1371%2Fjournal.pbio.1001908#s5

*Blot and Gel Data Policy*

Sincerely,

Richard

Richard Hodge, PhD

rhodge@plos.org

REVIEWS:

Reviewer #1 (Noboru Mizushima, signs review): The authors conducted an imaging-based screen to identify ER-phagy factors and discovered a REEP-like protein named Yep1. Yep1 plays a specific role in ER-phagy and nucleophagy at the sequestration step but not the initiation step. Yep1 exhibits membrane-shaping activity and requires its oligomerization property and the C-terminal amphipathic helices for ER-phagy. Structurally similar mammalian REEP proteins and the budding yeast Atg40 (even lacking the AIM motif) can rescue the loss of Yep1. Atg40 in budding yeast is, therefore, like a hybrid molecule of Yep1 and Epr1 in fission yeast.

This study significantly contributes to our understanding of the mechanism of ER-phagy in fission yeast. It also provides a clear explanation for the lack of membrane-shaping activity in the previously known ER-phagy receptor Epr1. The data presented are well-organized and compelling, and this paper is worth publishing. There are only a few minor comments to address.

1. The authors discuss two possibilities regarding the function of Yep1: (1) shaping of the cargo membrane and/or (2) shaping of the autophagosomal membrane. While it is the authors' prerogative to consider both possibilities, this reviewer finds the first hypothesis to be much more likely based on the findings in the present study. It might be beneficial for the authors to provide a bit stronger suggestion for the first hypothesis in the Abstract. Again, the decision is up to the authors.

2. The structures observed in DDT-treated yep1Δ cells are intriguing (Fig. 2B). However, it remains unclear why the formation of these structures is dependent on Atg5. This issue is not explained in the model presented in Fig. S7. It would be helpful to provide further explanation in the Discussion regarding ATG dependency. Furthermore, there are some related questions. Are the cargo structures that accumulate in yep1Δ cells larger than those observed in autophagosomes in wild-type cells? Are these structures completely detached from the source membranes (no minor connections)? Discussing these points would be helpful for readers.

3. Recently, a highly relevant paper was published by Kanki's group (PMID: 37191320). Please cite and discuss this paper.

Reviewer #2: In the manuscript, the authors identify the protein Yep1, the yeast ortholog of mammalian REEPs, as a component of the ER-phagy/nucleophagy machinery. Yep1 is not affecting the initiation of the autophagic process but is rather involved in the enclosure of the ER-phagy and nucleophagy cargos.

The ideas are interesting and the role of REEP proteins in autophagosome shaping and ER-phagy have been reported also by other groups (PMID: 37191320; doi.org/10.1101/2022.10.27.514035). The fact that two additional publications reported similar results is not really affecting the novelty. One paper has been published very recently and the other one is deposited in bioRxiv. 

The main point with the manuscript is a lack of experimental context. The manuscript a bit confusing, the final message, that the authors want to deliver, is not clear. While the role of Yep1 is presented as crucial for cargos delivery to vacuole, on the other side the Atg40 protein seems to fully compensate for the absence of Yep1. Therefore, it is not clear the relation between Atg40 and Yep1. Despite the structural similarities regarding the amphipathic helixes, the two proteins are distinct. Further investigations are needed to establish the clear relation between the ER-phagy receptor Atg40 and ER structural protein Yep1. Do they directly interact? Are they involved in the formation of some protein clusters that functionally connect these proteins during the ER-phagy process? 

Moreover, many experiments are redundant and recapitulate already published data (all the experiments made with Rtn1 mutants are not really necessary and already reported in literature). Most of the experiments are based on yeast genetics, that is good, but the molecular mechanisms are also quite speculative. How Yep1 function is regulated? Is it part of a larger protein complex? Is it post-translationally modified? 

This manuscript has a potentially solid idea, but it needs quite a lot of improvements regarding the molecular mechanisms that are needed to support authors' hypothesis. 

Authors should also consider including in the manuscript, maybe in the discussion the recent published literature on ER-phagy.

Reviewer #3: In this study, Zou et al. identified that the REEP family protein Yep1 as a novel factor involved in ER-phagy and nucleophagy in Schizosaccharomyces pombe. In cells lacking Yep1, ER-phagy and nucleophagy are completely blocked, but bulk autophagy is almost normal. Yep1 is not required for the initial step of ER-phagy in which the ER-phagy receptor Epr1 and Atg8 are colocalized. Based on the results of fluorescence microscopy and electron microscopy with the genetically encoded EM tag MTn, ring-shaped membrane structures derived from nucleus or ER were shown to accumulate in yep1Δ cells. Yep1, like known ER-shaping proteins such as Yop1 and Rtn1, has the ability to self-interact and promote ER morphogenesis. The authors showed that self-interaction and the C-terminal amphipathic helices of Yep1 are required for ER-phagy. The authors also showed that human REEP1-4 and budding yeast Atg40 can rescue the defect of ER-phagy in yep1Δ cells. Given these results, the authors propose that Yep1 functions in the autophagosomal enclosure of nucleus/ER derived membrane structures. The finding in this article is novel and important for understanding how the substrates in the ER and nucleus are sequestered into autophagosomes, but the authors need to address the following issues to improve the manuscript.

1. In Figure 1H and I, the authors stated that the deletion of EPR1 affected the degradation of Pus1-mCitrine in DTT-treated cells more severely than that in nitrogen starved cells. However, it seems that cleaved mCitrine was not detected in epr1Δ cells just because it was less produced even in wild-type cells treated with DTT rather than because ERP1 deletion more strongly impaired nucleophagy induced by DTT treatment.

2. When ER-phagy and nucleophagy are induced, does Yep1 accumulate in the site of the ER/nucleus derived vesicles formation? The authors only show the localization of Yep1 during the logarithmic growth phase. The authors should observe the localization of Yep1 when ER-phagy and nucleophagy are induced.

3. In Figure S1E, the number of double ring structures in cells treated with DTT is larger than that in nitrogen starved cells. This seems inconsistent with the results of the GFP cleavage assay (Figure 1F, G). The authors should mention this apparent discrepancy.

4. Based on the results of electron microscopic analysis, the authors claimed that there is no isolation membrane around nucleus- or ER-derived structures. The authors should also confirm this with fluorescence microscopy analysis (colocalization of Atg8 and cytoplasmic Bqt4 (or Pus1) puncta in yep1Δ cells after 12 h DTT treatment.)

5. The authors should check the levels of proteins expressed from plasmids in Figs. 3K, M, S3F, S5A, 4C, E, G, H, S6D, E, G, H.

6. In Figure S4, some labels are missing.

Reviewer #4: The degradation of ER by autophagy is termed ER-phagy. In order to identify novel factors required for ER-phagy in fission yeast, the authors performed a chemical mutagenesis screen. They screened for mutants which failed to transport the ER membrane protein Ost4-GFP to the vacuole. From this screen they identified a previously uncharacterized protein which they called Yep1. The authors confirm that deletion of yep1 in S. pombe leads to a loss of ER-phagy in response to both DTT and -N. They further demonstrate that yep1 is required for nucleophagy (autophagy of the nucleus) but it is not required for bulk (non-selective) autophagy. Interestingly, the authors demonstrate that Yep1 does not function as a selective autophagy receptor as it can not interact with Atg8 and that it likely contains a different function. EM imaging suggested that stalled ER-phagy cargos in yep1delta cells are not enclosed in autophagosomes. To verify this they use an auxin inducible degradation system to demonstrate that ER-phagy cargos are accessible to the cytoplasm and degraded in -N media. They next demonstrate that deletion of yep1 alters ER morphology suggesting that yep1 plays a role in ER shaping. They demonstrate that yep1 can oligomerize (likely a dimer) by co-IP and BIFC. They demonstrate that Yep1 shares similar features with the REEP family of proteins and Atg40 (an ER-phagy receptor in S. cerevisiae). They demonstrate that REEP1-4 and Atg40 can rescue the ER-phagy defect of yep1 suggesting that these proteins have similar roles in ER-phagy. Taken together all of their data suggests one of two possible roles for yep1 in ER-phagy and nucleophagy. The first is that yep1 plays a role in shaping cargo membranes while in the second it may play a role in shaping the autophagosomal membrane. 

This is a very interesting paper with new and exciting findings about the mechanisms of autophagy. The paper is well written and the conclusions are well supported. The authors were careful to use complimentary assays (co-IP and BIFC or light microscopy and EM) to strengthen their findings. They authors also make good use of controls for the majority of their assays. It is also nice to see the honesty and openness of the discussion section. The authors are careful to state different possibilities for their data. They also carefully compare their data to another recent manuscript without criticizing the other paper. One major weakness of the manuscript is the lack of quantification of their ER-phagy, nucleophagy and bulk autophagy assays. The authors have many instances where a single western blot is shown for each experiment. Repeating these assays in triplicate and quantification of these blots would further strengthen their findings. 

Major concerns 

1. The authors rely heavily on ER-phagy, nucleophagy and bulk autophagy assays which show the generation of free GFP from a protein that is captured in these processes. However, this results in individual western blots where only single points of data are shown. The authors should repeat these results in triplicate for all of these experiments in the main figures and quantify them to strengthen their claims. The authors should repeat and quantify Figure 1D, 1E, 1F, 1G, 1H, 1I, 1J, 3J, 3K, 3M 4C, 4E, 4G, 4H. 

2. It is not entirely clear how the authors noted the formation of the Bqt4 and Pus1 puncta that are utilized in Figure 2. It would be helpful if this could be explained in more detail in the manuscript. 

3. In the CO-IP experiments in Figure 3D and 3H, why does Yep1-GFP appear as a doublet? If the 34 kDa band is largely free GFP or mcherry why does this pulldown with the sample? Do the authors have a hypothesis for this doublet. 

 4. In Figure 3F the overall structure is hard to compare to 3G. It may be helpful to color one of the monomers in 3F grey and the other monomer the colors used in 3G. This would help connect the two figures and make it clear which helices are the APHs to be discussed later. 

5. The authors mention that mutation of the APH1 helix did not abolish ER-phagy but it did lead to a dramatic reduction in ER-phagy in Figure 3M. The authors should acknowledge this in the results.

---

## [Editor Report · Decision Letter 2]

3 Oct 2023

Dear Dr Du,

Thank you for your patience while we considered your revised manuscript "Fission yeast ortholog of human REEP1-4 is required for autophagosomal enclosure of ER-phagy/nucleophagy cargos" for publication as a Short Report at PLOS Biology. This revised version of your manuscript has been evaluated by the PLOS Biology editors, the Academic Editor.

Based on our Academic Editor's assessment of your revision, I am pleased to say that we are likely to accept this manuscript for publication, provided you satisfactorily address the following data and other policy-related requests that I have provided below:

(A) We would like to suggest the following modification to the title: 

"The ortholog of human REEP1-4 is required for autophagosomal enclosure of ER-phagy/nucleophagy cargos in fission yeast"

(B) You may be aware of the PLOS Data Policy, which requires that all data be made available without restriction: http://journals.plos.org/plosbiology/s/data-availability. For more information, please also see this editorial: http://dx.doi.org/10.1371/journal.pbio.1001797

-Supplementary files (e.g., excel). Please ensure that all data files are uploaded as 'Supporting Information' and are invariably referred to (in the manuscript, figure legends, and the Description field when uploading your files) using the following format verbatim: S1 Data, S2 Data, etc. Multiple panels of a single or even several figures can be included as multiple sheets in one excel file that is saved using exactly the following convention: S1_Data.xlsx (using an underscore).

-Deposition in a publicly available repository. Please also provide the accession code or a reviewer link so that we may view your data before publication. 

Figure 1D-J, 2I-J, 3C, 3H, 3J-L, 4C, 4F, 4H-I, S1A, S1D, S1G, S2B-D, S2F-G, S4A-D, S5B-C, S6D

(C) Please also ensure that each of the relevant figure legends in your manuscript include information on *WHERE THE UNDERLYING DATA CAN BE FOUND*, and ensure your supplemental data file/s has a legend.

(D) We require the original, uncropped and minimally adjusted images supporting all blot and gel results reported in the following Figures:

Figure 1D-J, 3D, 3I-L, 4C, 4F, 4H-I, S1E, S1I, S1K-L, S4E-F, S5E-H, S6A-C, S7D-E, S7G-H

We will require these files before a manuscript can be accepted so please prepare and upload them now. Please carefully read our guidelines for how to prepare and upload this data: https://journals.plos.org/plosbiology/s/figures#loc-blot-and-gel-reporting-requirements

(E) Please ensure that your Data Statement in the submission system accurately describes where your data can be found and is in final format, as it will be published as written there. 

(F) Please note that per journal policy, the model system/species studied should be clearly stated in the abstract of your manuscript. 

(G) Please also provide a blurb which (if accepted) will be included in our weekly and monthly Electronic Table of Contents, sent out to readers of PLOS Biology, and may be used to promote your article in social media. The blurb should be about 30-40 words long and is subject to editorial changes. It should, without exaggeration, entice people to read your manuscript. It should not be redundant with the title and should not contain acronyms or abbreviations. For examples, view our author guidelines: https://journals.plos.org/plosbiology/s/revising-your-manuscript#loc-blurb

We expect to receive your revised manuscript within two weeks. 

*Published Peer Review History*

*Press*

Kind regards,

Richard

Richard Hodge, PhD

rhodge@plos.org

---

## [Editor Report · Decision Letter 3]

10 Oct 2023

Dear Dr Du,

On behalf of my colleagues and the Academic Editor, Sharon Tooze, I am pleased to say that we can accept your manuscript for publication, provided you address any remaining formatting and reporting issues. These will be detailed in an email you should receive within 2-3 business days from our colleagues in the journal operations team; no action is required from you until then. Please note that we will not be able to formally accept your manuscript and schedule it for publication until you have completed any requested changes.

PRESS

Best wishes, 

Richard

Richard Hodge, PhD

rhodge@plos.org

PLOS
